# Finite-Sample Analysis of Off-Policy TD-Learning via Generalized Bellman Operators

**Zaiwei Chen**
Georgia Institute of Technology

**Siva Theja Maguluri**
Georgia Institute of Technology

**Sanjay Shakkottai**
The University of Texas at Austin

**Karthikeyan Shanmugam**
IBM Research NY

## Abstract

In TD-learning, off-policy sampling is known to be more practical than on-policy sampling, and by decoupling learning from data collection, it enables data reuse. It is known that policy evaluation has the interpretation of solving a generalized Bellman equation. In this paper, we derive finite-sample bounds for any general off-policy TD-like stochastic approximation algorithm that solves for the fixed-point of this generalized Bellman operator. Our key step is to show that the generalized Bellman operator is simultaneously a contraction mapping with respect to a weighted $\ell_p$-norm for each $p$ in $[1, \infty)$, with a common contraction factor. Off-policy TD-learning is known to suffer from high variance due to the product of importance sampling ratios. A number of algorithms (e.g. $Q^\pi(\lambda)$, Tree-Backup$(\lambda)$, Retrace$(\lambda)$, and $Q$-trace) have been proposed in the literature to address this issue. Our results immediately imply finite-sample bounds of these algorithms. In particular, we provide first-known finite-sample guarantees for $Q^\pi(\lambda)$, Tree-Backup$(\lambda)$, and Retrace$(\lambda)$, and improve the best known bounds of $Q$-trace in [19]. Moreover, we show the bias-variance trade-offs in each of these algorithms.

## 1 Introduction

Reinforcement learning (RL) demonstrated its success in learning effective policies for a variety of decision making problems such as autonomous driving [25, 26], recommender systems [1, 41], and game-related problems [23, 27, 39]. In RL, there is an important sub-problem – called the policy evaluation problem – of estimating the expected long term reward of a given policy. Solving the policy evaluation problem is usually an itermediate step in many existing RL algorithms to ultimately find an optimal policy, such as approximate policy iteration and actor-critic framework.

The policy evaluation problem is usually solved with the TD-learning method [30]. A key ingredient in TD-learning is the policy used to collect samples (called the behavior policy). Ideally, we want to generate samples from the target policy whose value function we want to estimate, and this is called on-policy sampling. However, in many cases such on-policy sampling is not possible due to practical reasons [16, 40], and hence we need to work with historical data that is generated by a possibly different policy (i.e., off-policy sampling). Although off-policy sampling is more practical than on-policy sampling, it is more challenging to analyze and is known to have high variance [15], which is a fundamental difficulty in off-policy learning. To overcome this difficulty, many variants of off-policy TD-learning algorithms have been proposed in the literature, such as $Q^\pi(\lambda)$ [17], Tree-Backup$(\lambda)$ (henceforth denoted by TB$(\lambda)$) [24], Retrace$(\lambda)$ [22], and $Q$-trace [19], etc.

### 1.1 Main Contributions

In this work, we establish finite-sample bounds of a general $n$-step off-policy TD-learning algorithm that also subsumes several algorithms presented in the literature. The key step is to show that such

35th Conference on Neural Information Processing Systems (NeurIPS 2021)

algorithm can be modeled as a Markovian stochastic approximation (SA) algorithm for solving a generalized Bellman equation. We present sufficient conditions under which the generalized Bellman operator is contractive with respect to a weighted $\ell_p$-norm for every $p \in [1, \infty)$, with a uniform contraction factor for all $p$. Our result shows that the sample complexity scales as $\tilde{\mathcal{O}}(\epsilon^{-2})$, where $\epsilon$ is the required accuracy. It also involves a factor that depends on the problem parameters, in particular, the generalized importance sampling ratios, and explicitly demonstrates the bias-variance trade-off.

Our result immediately gives finite-sample guarantees for variants of multi-step off-policy TD-learning algorithms including $Q^\pi(\lambda)$, TB$(\lambda)$, Retrace$(\lambda)$, and $Q$-trace. For $Q^\pi(\lambda)$, TB$(\lambda)$, and Retrace$(\lambda)$, we establish the first-known results in the literature, while for $Q$-trace, we improve the best known results in [19] in terms of the dependency on the size of the state-action space. The weighted $\ell_p$-norm contraction property with a uniform contraction factor for all $p \in [1, \infty)$ is crucial for us to establish the improved sample complexity. Based on the finite-sample bounds, we show that all four algorithms overcome the high variance issue in Vanilla off-policy TD-learning, but their convergence rates are all affected to varying degrees.

## 1.2 Generalized Bellman Operator and Stochastic Approximation

In this section, we illustrate the interpretation of off-policy multi-step TD-learning as an SA algorithm for solving a generalized Bellman equation. Consider the policy evaluation problem where the goal is to estimate the state-action value function $Q^\pi$ of a given policy $\pi$. In the simplest setting where TD(0) with on-policy sampling is employed, it is well known that the algorithm is an SA algorithm for solving the Bellman equation $Q = \mathcal{H}_\pi(Q)$, where $\mathcal{H}_\pi(\cdot)$ is the Bellman operator. The generalized Bellman operator $\mathcal{B}(\cdot)$ we consider in this paper is defined by:

$$\mathcal{B}(Q) = \mathcal{T}(\mathcal{H}(Q) - Q) + Q, \tag{1}$$

where $\mathcal{T}(\cdot)$ and $\mathcal{H}(\cdot)$ are two auxiliary operators. In the special case where $\mathcal{T}(\cdot) = I(\cdot)$ and $\mathcal{H}(\cdot) = \mathcal{H}_\pi(\cdot)$, the generalized Bellman operator $\mathcal{B}(\cdot)$ reduces to the regular Bellman operator $\mathcal{H}_\pi(\cdot)$. Note that any fixed point of $\mathcal{H}(\cdot)$ is also a fixed point of $\mathcal{B}(\cdot)$, as long as $\mathcal{T}(\cdot)$ is such that $\mathcal{T}(\mathbf{0}) = \mathbf{0}$. Thus, the operator $\mathcal{H}(\cdot)$ controls the fixed-point of the generalized Bellman operator $\mathcal{B}(\cdot)$, and as we will see later, the operator $\mathcal{T}(\cdot)$ can be used to control its contraction properties.

To further understand the operator $\mathcal{B}(\cdot)$, we demonstrate in the following that both on-policy $n$-step TD and TD$(\lambda)$ can be viewed as SA algorithms for solving the generalized Bellman equation $\mathcal{B}(Q) = Q$, with different auxiliary operators $\mathcal{T}(\cdot)$ and $\mathcal{H}(\cdot)$. On-policy $n$-step TD is designed to solve the $n$-step Bellman equation $Q = \mathcal{H}_\pi^n(Q)$, which can be explicitly written as $Q = \sum_{i=0}^{n-1} (\gamma P_\pi)^i R + (\gamma P_\pi)^n Q$. Here $R$ is the reward vector, $\gamma$ is the discount factor, and $P_\pi$ is the transition probability matrix under policy $\pi$. By reverse telescoping, the $n$-step Bellman equation is equivalent to $Q = \sum_{i=0}^{n-1} (\gamma P_\pi)^i (R + \gamma P_\pi Q - Q) + Q = \mathcal{T}(\mathcal{H}_\pi(Q) - Q) + Q$, where $\mathcal{T}(Q) = \sum_{i=0}^{n-1} (\gamma P_\pi)^i Q$. Similarly, one can formulate the TD$(\lambda)$ Bellman equation in the form of $\mathcal{B}(Q) = Q$, where $\mathcal{T}(Q) = (1-\lambda) \sum_{i=0}^\infty \lambda^i \sum_{j=0}^{i-1} (\gamma P_\pi)^i Q$ and $\mathcal{H}(\cdot) = \mathcal{H}_\pi(\cdot)$.

In these examples, the operator $\mathcal{T}(\cdot)$ determines the contraction factor of $\mathcal{B}(\cdot)$ by controlling the degree of bootstrapping. In this work, we show that in addition to on-policy TD-learning, variants of off-policy TD-learning with multi-step bootstrapping and generalized importance sampling ratios can also be interpreted as SA algorithms for solving the generalized Bellman equation. Moreover, under some mild conditions, we show that the generalized Bellman operator $\mathcal{B}(\cdot)$ is a contraction mapping with respect to some weighted $\ell_p$-norm for any $p \in [1, \infty)$, with a common contraction factor. This enables us to establish finite-sample bounds of general multi-step off-policy TD-like algorithms.

## 1.3 Related Literature

The TD-learning method was first proposed in [30] for solving the policy evaluation problem. Since then, there is an increasing interest in theoretically understanding TD-learning and its variants.

*On-Policy TD-Learning.* The most basic TD-learning method is the TD(0) algorithm [30]. Later it was extended to using multi-step bootstrapping (i.e., the $n$-step TD-learning algorithm [11, 37, 38]), and using eligibility trace (i.e., the TD$(\lambda)$ algorithm [28, 30]). The asymptotic convergence of TD-learning was established in [13, 18, 35]. As for finite-sample analysis, a unified Lyapunov approach is presented in [10]. To overcome the curse of dimensionality in RL, TD-learning is usually incorporated with function approximation in practice. In the basic setting where a linear parametric architecture is

used, the asymptotic convergence of TD-learning was established in [36], and finite-sample bounds in [5, 12, 29, 34]. Very recently, the convergence and finite-sample guarantee of TD-learning with neural network approximation were studied in [7, 8].

*Off-Policy TD-Learning.* In the off-policy setting, since the samples are not necessarily generated by the target policy, usually importance sampling ratios (or "generalized" importance sampling ratios) are introduced in the TD-learning algorithm. The resulting algorithms are $Q^\pi(\lambda)$ [24], TB$(\lambda)$ [17], Retrace$(\lambda)$ [22], and $Q$-trace [19] (which is an extension of $V$-trace [14]), etc. The asymptotic convergence of these algorithms has been established in the papers in which they were proposed. To the best of our knowledge, finite-sample guarantees are established only for $Q$-trace and $V$-trace [9, 10, 19]. In the function approximation setting, TD-learning with off-policy sampling and function approximation is a typical example of the deadly triad [31], and can be unstable [2, 31]. To achieve convergence, one needs to significantly modify the original TD-learning algorithm, resulting in two time-scale algorithms such as GTD [21], TDC [32], and emphatic TD [33], etc.

### 1.4 Preliminaries

In this section, we cover the background of RL and the TD-learning method for solving the policy evaluation problem. The RL problem is usually modeled as a Markov decision process (MDP). In this work, we consider an MDP with a finite set of states $\mathcal{S}$, a finite set of actions $\mathcal{A}$, a set of unknown action dependent transition probability matrices $\mathcal{P} = \{P_a \in \mathbb{R}^{|\mathcal{S}| \times |\mathcal{S}|} \mid a \in \mathcal{A}\}$, an unknown reward function $\mathcal{R} : \mathcal{S} \times \mathcal{A} \mapsto [0, 1]$, and a discount factor $\gamma \in (0, 1)$. In order for an MDP to progress, we must specify the policy of selecting actions based on the state of the environment. Specifically, a policy $\pi$ is a mapping from the state-space to probability distributions supported on the action space, i.e., $\pi : \mathcal{S} \mapsto \Delta^{|\mathcal{A}|}$. The state-action value function $Q^\pi$ associated with a policy $\pi$ is defined by $Q^\pi(s, a) = \mathbb{E}_\pi[\sum_{k=0}^\infty \gamma^k \mathcal{R}(S_k, A_k) \mid S_0 = s, A_0 = a]$ for all $(s, a)$. The goal in policy evaluation is to estimate the state-action value function $Q^\pi$ for a given policy $\pi$.

Since the transition probabilities as well as the reward function are unknown, such state-action value function cannot be directly computed. The TD-learning algorithm is designed to estimate $Q^\pi$ using the SA method. Specifically, in TD-learning, we first collect a sequence of samples $\{(S_k, A_k)\}$ from the model using some behavior policy $\pi_b$. Then the value function $Q^\pi$ is iteratively estimated using the samples $\{(S_k, A_k)\}$. When $\pi_b = \pi$, the algorithm is called on-policy TD-learning, otherwise the algorithm is referred to as off-policy TD-learning.

## 2 Finite-Sample Analysis of General Off-Policy TD-Learning

In this section, we present our unified framework for finite-sample analysis of off-policy TD-learning algorithms using generalized importance sampling ratios and multi-step bootstrapping. The proofs of all technical results presented in this paper are provided in the Appendix.

### 2.1 A Generic Model for Multi-Step Off-Policy TD-Learning

Algorithm 1 presents our generic algorithm model. Due to off-policy sampling, the two functions $c, \rho : \mathcal{S} \times \mathcal{A} \mapsto \mathbb{R}_+$ are introduced in Algorithm 1 to serve as generalized importance sampling ratios in order to account for the discrepancy between the target policy $\pi$ and the behavior policy $\pi_b$. We denote $c_{\max} = \max_{s,a} c(s, a)$ and $\rho_{\max} = \max_{s,a} \rho(s, a)$. We next show how Algorithm 1 captures variants of off-policy TD-learning algorithms in the literature by using different generalized importance sampling ratios $c(\cdot, \cdot)$ and $\rho(\cdot, \cdot)$.

*Vanilla IS.* When $c(s, a) = \rho(s, a) = \pi(a|s)/\pi_b(a|s)$ for all $(s, a)$, Algorithm 1 is the standard off-policy TD-learning with importance sampling [24]. We will refer to this algorithm as Vanilla IS. Although Vanilla IS was shown to converge to $Q^\pi$ [24], since the product of importance sampling ratios $\prod_{j=k+1}^i \frac{\pi(A_j|S_j)}{\pi_b(A_j|S_j)}$ is not controlled in any way, it suffers the most from high variance.

*The $Q^\pi(\lambda)$ Algorithm.* When $c(s, a) = \lambda$ and $\rho(s, a) = \pi(a|s)/\pi_b(a|s)$, Algorithm 1 is the $Q^\pi(\lambda)$ algorithm [17]. The $Q^\pi(\lambda)$ algorithm overcomes the high variance issue in Vanilla IS by introducing the parameter $\lambda$. However, the algorithm converges only when $\lambda$ is sufficiently small [22].

*The TB$(\lambda)$ Algorithm.* When $c(s, a) = \lambda\pi(a|s)$ and $\rho(s, a) = \pi(a|s)/\pi_b(a|s)$, we have the TB$(\lambda)$ algorithm [24]. The TB$(\lambda)$ algorithm also overcomes the high variance issue in Vanilla IS and is

---

**Algorithm 1** A Generic Algorithm for Multi-Step Off-Policy TD-Learning

---

1: **Input:** $K$, $\{\alpha_k\}$, $Q_0$, $\pi$, $\pi_b$, generalized importance sampling ratios $c, \rho : \mathcal{S} \times \mathcal{A} \mapsto \mathbb{R}_+$, and sample trajectory $\{(S_k, A_k)\}_{0 \leq k \leq K+n}$ collected under the behavior policy $\pi_b$.
2: **for** $k = 0, 1, \cdots, K - 1$ **do**
3:     $\alpha_k(s, a) = \alpha_k \mathbb{I}\{(s, a) = (S_k, A_k)\}$ for all $(s, a)$
4:     $\Delta(S_i, A_i, S_{i+1}, A_{i+1}, Q_k) = \mathcal{R}(S_i, A_i) + \gamma \rho(S_{i+1}, A_{i+1}) Q_k(S_{i+1}, A_{i+1}) - Q_k(S_i, A_i)$ for all $i \in \{k, k+1, ..., k+n-1\}$.
5:     $Q_{k+1}(s, a) = Q_k(s, a) + \alpha_k(s, a) \sum_{i=k}^{k+n-1} \gamma^{i-k} \prod_{j=k+1}^{i} c(S_j, A_j) \Delta(S_i, A_i, S_{i+1}, A_{i+1}, Q_k)$ for all $(s, a)$
6: **end for**
7: **Output:** $Q_K$

---

guaranteed to converge to $Q^\pi$ without needing any strong assumptions. However, as discussed in [22], the TB($\lambda$) algorithm lacks sample efficiency as it does not effectively use the multi-step return.

*The Retrace($\lambda$) Algorithm.* When $c(s, a) = \lambda \min(1, \pi(a|s)/\pi_b(a|s))$ and $\rho(s, a) = \pi(a|s)/\pi_b(a|s)$, we have the Retrace($\lambda$) algorithm, which overcomes the high variance and converges to $Q^\pi$. The convergence rate of Retrace($\lambda$) is empirically observed to be better than TB($\lambda$) in [22].

*The Q-trace Algorithm.* When we choose $c(s, a) = \min(\bar{c}, \pi(a|s)/\pi_b(a|s))$ and $\rho(s, a) = \min(\bar{\rho}, \pi(a|s)/\pi_b(a|s))$, where $\bar{\rho} \geq \bar{c}$, Algorithm 1 is the Q-trace algorithm [19]. The Q-trace algorithm is an analog of the $V$-trace algorithm [14] in that Q-trace estimates the Q-function instead of the $V$-function. The two truncation levels $\bar{c}$ and $\bar{\rho}$ in these algorithms separately control the variance and the asymptotic bias in the algorithm respectively. Note that due to the truncation level $\bar{\rho}$, the algorithm no longer converges to $Q^\pi$, but to a biased limit point, denoted by $Q^{\pi,\rho}$ [19].

From now on, we focus on studying Algorithm 1. We make the following assumption about the behavior policy $\pi_b$, which is fairly standard in off-policy TD-learning.

**Assumption 2.1.** The behavior policy $\pi_b$ satisfies $\pi_b(a|s) > 0$ for all $(s, a)$. In addition, the Markov chain $\{S_k\}$ induced by the behavior policy $\pi_b$ is irreducible and aperiodic.

Irreducibility and aperiodicity together imply that the Markov chain $\{S_k\}$ has a unique stationary distribution, which we denote by $\kappa_S \in \Delta^{|\mathcal{S}|}$. Moreover, the Markov chain $\{S_k\}$ mixes geometrically fast in that there exist $C > 0$ and $\sigma \in (0, 1)$ such that $\max_{s \in \mathcal{S}} \|P^k(s, \cdot) - \kappa_S(\cdot)\|_{\text{TV}} \leq C\sigma^k$ for all $k \geq 0$, where $\| \cdot \|_{\text{TV}}$ is the total variation distance [20]. Let $\kappa_{SA} \in \Delta^{|\mathcal{S}||\mathcal{A}|}$ be such that $\kappa_{SA}(s, a) = \kappa_S(s)\pi_b(a|s)$ for all $(s, a)$. Note that $\kappa_{SA} \in \Delta^{|\mathcal{S}||\mathcal{A}|}$ is the stationary distribution of the Markov chain $\{(S_k, A_k)\}$ under the behavior policy $\pi_b$. Let $\mathcal{K}_S = \text{diag}(\kappa_S) \in \mathbb{R}^{|\mathcal{S}| \times |\mathcal{S}|}$, and let $\mathcal{K}_{SA} = \text{diag}(\kappa_{SA}) \in \mathbb{R}^{|\mathcal{S}||\mathcal{A}| \times |\mathcal{S}||\mathcal{A}|}$. Denote the minimal (maximal) diagonal entries of $\mathcal{K}_S$ and $\mathcal{K}_{SA}$ by $\mathcal{K}_{S,\min}$ ($\mathcal{K}_{S,\max}$) and $\mathcal{K}_{SA,\min}$ ($\mathcal{K}_{S,\max}$) respectively.

## 2.2 Identifying the Generalized Bellman Operator

In this section, we identify the generalized Bellman equation which Algorithm 1 is trying to solve, and also the corresponding generalized Bellman operator and its asynchronous variant. Let $\mathcal{T}_c, \mathcal{H}_\rho : \mathbb{R}^{|\mathcal{S}||\mathcal{A}|} \mapsto \mathbb{R}^{|\mathcal{S}||\mathcal{A}|}$ be two operators defined by

$$[\mathcal{T}_c(Q)](s, a) = \sum_{i=0}^{n-1} \gamma^i \mathbb{E}_{\pi_b} \left[ \prod_{j=1}^{i} c(S_j, A_j) Q(S_i, A_i) \mid S_0 = s, A_0 = a \right], \quad \text{and}$$

$$[\mathcal{H}_\rho(Q)](s, a) = \mathcal{R}(s, a) + \gamma \mathbb{E}_{\pi_b} [\rho(S_{k+1}, A_{k+1}) Q(S_{k+1}, A_{k+1}) \mid S_k = s, A_k = a]$$

for all $(s, a)$. Note that the operator $\mathcal{T}_c(\cdot)$ depends on the generalized importance sampling ratio $c(\cdot, \cdot)$, while the operator $\mathcal{H}_\rho(\cdot)$ depends on the generalized importance sampling ratio $\rho(\cdot, \cdot)$.

With $\mathcal{T}_c(\cdot)$ and $\mathcal{H}_\rho(\cdot)$ defined above, Algorithm 1 can be viewed as an asynchronous SA algorithm for solving the generalized Bellman equation $\mathcal{B}_{c,\rho}(Q) = Q$, where the generalized Bellman operator $\mathcal{B}_{c,\rho}(\cdot)$ is defined by $\mathcal{B}_{c,\rho}(Q) = \mathcal{T}_c(\mathcal{H}_\rho(Q) - Q) + Q$. Since Algorithm 1 performs asynchronous update, using the terminology in [10], we further define the asynchronous variant $\tilde{\mathcal{B}}_{c,\rho}(\cdot)$ of the generalized Bellman operator $\mathcal{B}_{c,\rho}(\cdot)$ by

$$\tilde{\mathcal{B}}_{c,\rho}(Q) := \mathcal{K}_{SA}\mathcal{B}_{c,\rho}(Q) + (I - \mathcal{K}_{SA})Q = \mathcal{K}_{SA}\mathcal{T}_c(\mathcal{H}_\rho(Q) - Q) + Q. \tag{2}$$

Each component of the asynchronous generalized Bellman operator $\tilde{\mathcal{B}}_{c,\rho}(\cdot)$ can be thought of as a convex combination with identity, where the weights are the stationary probabilities of visiting state-action pairs. This captures the fact that when performing asynchronous update, the corresponding component is updated only when the state-action pair is visited. It is clear from its definition that $\tilde{\mathcal{B}}_{c,\rho}(\cdot)$ has the same fixed-points as $\mathcal{B}_{c,\rho}(\cdot)$ (provided that they exist). See [10] for a more detailed explanation about asynchronous Bellman operators.

Under some mild conditions on the generalized importance sampling ratios $c(\cdot,\cdot)$ and $\rho(\cdot,\cdot)$, we will show in the next section that both the asynchronous generalized Bellman operator $\tilde{\mathcal{B}}_{c,\rho}(\cdot)$ and the operator $\mathcal{H}_\rho(\cdot)$ are contraction mappings. Therefore, since $\mathcal{T}_c(\mathbf{0}) = \mathbf{0}$, the operators $\mathcal{H}_\rho(\cdot)$, $\mathcal{B}_{c,\rho}(\cdot)$, $\tilde{\mathcal{B}}_{c,\rho}(\cdot)$ all share the same unique fixed-point. Since the fixed-point of the operator $\mathcal{H}_\rho(\cdot)$ depends only on the generalized importance sampling ratio $\rho(\cdot,\cdot)$, but not on $c(\cdot,\cdot)$, we can flexibly choose $c(\cdot,\cdot)$ to control the variance while maintaining the fixed-point of the operator $\tilde{\mathcal{B}}_{c,\rho}(\cdot)$. As we will see later, this is the key property used in designing variants of variance reduced $n$-step off-policy TD-learning algorithms such as $Q^\pi(\lambda)$, TB($\lambda$), and Retrace($\lambda$).

## 2.3 Establishing the Contraction Property

In this section, we study the fixed-point and the contraction property of the asynchronous generalized Bellman operator $\tilde{\mathcal{B}}_{c,\rho}(\cdot)$. We begin by introducing some notation. Let $D_c, D_\rho \in \mathbb{R}^{|\mathcal{S}||\mathcal{A}| \times |\mathcal{S}||\mathcal{A}|}$ be two diagonal matrices such that $D_c((s,a),(s,a)) = \sum_{a' \in \mathcal{A}} \pi_b(a'|s)c(s,a')$ and $D_\rho((s,a),(s,a)) = \sum_{a' \in \mathcal{A}} \pi_b(a'|s)\rho(s,a')$ for all $(s,a)$. We denote $D_{c,\min}$ $(D_{c,\max})$ and $D_{\rho,\min}$ $(D_{\rho,\max})$ as the minimal (maximal) diagonal entries of the matrices $D_c$ and $D_\rho$ respectively.

In view of the definition of $\tilde{\mathcal{B}}_{c,\rho}(\cdot)$ in Eq. (2), any fixed-point of $\mathcal{H}_\rho(\cdot)$ must also be a fixed-point of $\tilde{\mathcal{B}}_{c,\rho}(\cdot)$. We first study the fixed point of $\mathcal{H}_\rho(\cdot)$ by establishing its contraction property.

**Proposition 2.1.** *Suppose that $D_{\rho,\max} < 1/\gamma$. Then the operator $\mathcal{H}_\rho(\cdot)$ is a contraction mapping with respect to the $\ell_\infty$-norm, with contraction factor $\gamma D_{\rho,\max}$. In this case, the fixed-point $Q^{\pi,\rho}$ of $\mathcal{H}_\rho(\cdot)$ satisfies the following two inequalities: (1) $\|Q^\pi - Q^{\pi,\rho}\|_\infty \leq \frac{\gamma \max_{s \in \mathcal{S}} \sum_{a \in \mathcal{A}} |\pi(a|s) - \pi_b(a|s)\rho(s,a)|}{(1-\gamma)(1-\gamma D_{\rho,\max})}$, and (2) $\|Q^{\pi,\rho}\|_\infty \leq \frac{1}{1-\gamma D_{\rho,\max}}$.*

Observe from Proposition 2.1 (1) that when $\rho(s,a) = \pi(a|s)/\pi_b(a|s)$, which is the case for $Q^\pi(\lambda)$, TB($\lambda$), and Retrace($\lambda$), the unique fixed-point $Q^{\pi,\rho}$ is exactly the target value function $Q^\pi$. This agrees with the definition of the operator $\mathcal{H}_\rho(\cdot)$ in that it reduces to the regular Bellman operator $\mathcal{H}_\pi(\cdot)$ when $\rho(s,a) = \pi(a|s)/\pi_b(a|s)$ for all $(s,a)$. If $\rho(s,a) \neq \pi(a|s)/\pi_b(a|s)$ for some $(s,a)$, then in general the fixed-point of $\mathcal{H}_\rho(\cdot)$ is different from $Q^\pi$. See Appendix A.2 for more details. In that case, Proposition 2.1 provides an error bound on the difference between the potentially biased limit $Q^{\pi,\rho}$ and $Q^\pi$. Such error bound will be useful for us to study the $Q$-trace algorithm in Section 3. Proposition 2.1 (2) can be viewed as an analog to the inequality that $\|Q^\pi\|_\infty \leq 1/(1-\gamma)$ for any policy $\pi$. Since $\mathcal{H}_\rho(\cdot)$ is no longer the Bellman operator $\mathcal{H}_\pi(\cdot)$, the corresponding upper bound on the size of its fixed-point $Q^{\pi,\rho}$ also changes.

Note that Proposition 2.1 guarantees the existence and uniqueness of the fixed-point of the operator $\mathcal{H}_\rho(\cdot)$, hence also ensures the existence of fixed-points of the asynchronous generalized Bellman operator $\tilde{\mathcal{B}}_{c,\rho}(\cdot)$. To further guarantee the uniqueness of the fixed-point of $\tilde{\mathcal{B}}_{c,\rho}(\cdot)$, we establish its contraction property. We begin with the following definition.

**Definition 2.1.** *Let $\{\mu_i\}_{1 \leq i \leq d}$ be such that $\mu_i > 0$ for all $i$. Then for any $x \in \mathbb{R}^d$, the weighted $\ell_p$-norm ($p \in [1,\infty)$) of $x$ with weights $\{\mu_i\}$ is defined by $\|x\|_{\mu,p} = (\sum_i \mu_i |x_i|^p)^{1/p}$.*

We next establish the contraction property of the operator $\tilde{\mathcal{B}}_{c,\rho}(\cdot)$ in the following theorem. Let $\omega = \mathcal{K}_{SA,\min} f(\gamma D_{c,\min})(1 - \gamma D_{\rho,\max})$, where the function $f : \mathbb{R} \mapsto \mathbb{R}$ is defined by $f(x) = n$ when $x = 1$, and $f(x) = \frac{1-x^n}{1-x}$ when $x \neq 1$.

**Theorem 2.1.** *Suppose $c(s,a) \leq \rho(s,a)$ for all $(s,a)$ and $D_{\rho,\max} < 1/\gamma$. Then we have the following results: (1) For any $\theta \in (0,1)$, there exists a weight vector $\mu \in \Delta^{|\mathcal{S}||\mathcal{A}|}$ satisfying $\mu(s,a) \geq \frac{\omega(1-\theta)}{(1-\theta\omega)|\mathcal{S}||\mathcal{A}|}$ for all $(s,a)$ such that the operator $\tilde{\mathcal{B}}_{c,\rho}(\cdot)$ is a contraction mapping with respect to $\|\cdot\|_{\mu,p}$ for any $p \in [1,\infty)$, with contraction factor $\gamma_c = (1-\omega)^{1-1/p}(1-\theta\omega)^{1/p}$, (2) The operator $\tilde{\mathcal{B}}_{c,\rho}(\cdot)$ is a contraction mapping with respect to $\|\cdot\|_\infty$, with contraction factor $\gamma_c = 1 - \omega$.*

Consider Theorem 2.1 (1). Observe that we can further upper bound $\gamma_c = (1-\omega)^{1-1/p}(1-\theta\omega)^{1/p}$ by $1-\theta\omega$, which is independent of $p$ and is the uniform contraction factor we are going to use. Theorem 2.1 (2) can be viewed as an extension of Theorem 2.1 because $\lim_{p\to\infty}\|x\|_{\mu,p} = \|x\|_\infty$ for any $x \in \mathbb{R}^d$ and weight vector $\mu$, and $\lim_{p\to\infty}(1-\omega)^{1-1/p}(1-\theta\omega)^{1/p} = 1-\omega$.

Theorem 2.1 is the key result for our finite-sample analysis, and we present its proof in the next section. The weighted $\ell_p$-norm (especially the weighted $\ell_2$-norm) contraction property we established for the operator $\tilde{\mathcal{B}}_{c,\rho}(\cdot)$ has a far-reaching impact even beyond the finite-sample analysis of tabular RL in this paper. Specifically, recall that the key property used for establishing the convergence and finite-sample bound of on-policy TD-learning with *linear function approximation* in the seminal work [36] is that the corresponding Bellman operator is a contraction mapping not only with respect to the $\ell_\infty$-norm, but also with respect to a weighted $\ell_2$-norm. We establish the same property in the off-policy setting, and hence lay down the foundation for extending our results to the function approximation setting. This is an immediate future research direction.

## 2.4 Proof of Theorem 2.1

We begin by explicitly computing the asynchronous generalized Bellman operator $\tilde{\mathcal{B}}_{c,\rho}(\cdot)$. Let $\pi_c$ and $\pi_\rho$ be two policies defined by $\pi_c(a|s) = \frac{\pi_b(a|s)c(s,a)}{D_c((s,a),(s,a))}$ and $\pi_\rho(a|s) = \frac{\pi_b(a|s)\rho(s,a)}{D_\rho((s,a),(s,a))}$ for all $(s,a)$. Let $R \in \mathbb{R}^{|\mathcal{S}||\mathcal{A}|}$ be the reward vector defined by $R(s,a) = \mathcal{R}(s,a)$ for all $(s,a)$. For any policy $\pi'$, let $P_{\pi'}$ be the transition probability matrix of the Markov chain $\{(S_k, A_k)\}$ under $\pi'$, i.e., $P_{\pi'}((s,a),(s',a')) = P_a(s,s')\pi'(a'|s')$ for all state-action pairs $(s,a)$ and $(s',a')$.

**Proposition 2.2.** *The operator $\tilde{\mathcal{B}}_{c,\rho}(\cdot)$ is explicitly given by $\tilde{\mathcal{B}}_{c,\rho}(Q) = AQ + b$, where $A = I - \mathcal{K}_{SA}\sum_{i=0}^{n-1}(\gamma P_{\pi_c}D_c)^i(I - \gamma P_{\pi_\rho}D_\rho)$ and $b = \mathcal{K}_{SA}\sum_{i=0}^{n-1}(\gamma P_{\pi_c}D_c)^iR$.*

In light of Proposition 2.2, to prove Theorem 2.1, it is enough to study the matrix $A$. To proceed, we require the following definition.

**Definition 2.2.** Given $\beta \in [0,1]$, a matrix $M \in \mathbb{R}^{d\times d}$ is called a substochastic matrix with modulus $\beta$ if and only if $M_{ij} \geq 0$ for all $i,j$ and $\sum_j M_{ij} \leq 1-\beta$ for all $i$.

*Remark.* Note that for any non-negative matrix $M$, we have $\|M\|_\infty = \max_i \sum_j M_{ij}$. Therefore, a matrix $M$ being a substochastic matrix with modulus $\beta$ automatically implies that $\|M\|_\infty \leq 1-\beta$.

We next show in the following two propositions that (1) the matrix $A$ given in Proposition 2.2 is a substochastic matrix with modulus $\omega$, and (2) for any substochastic matrix $M$ with a positive modulus, there exist weights $\{\mu_i\}$ such that the induced matrix norm $\|M\|_{\mu,p}$ is strictly less than 1. These two results together immediately imply Theorem 2.1.

**Proposition 2.3.** *Suppose that $c(s,a) \leq \rho(s,a)$ for all $(s,a)$ and $D_{\rho,\max} < 1/\gamma$. Then the matrix $A$ given in Proposition 2.2 is a substochastic matrix with modulus $\omega$.*

The condition $c(s,a) \leq \rho(s,a)$ ensures that the matrix $A$ is non-negative, and the condition $D_{\rho,\max} < 1/\gamma$ ensures that the each row of the matrix $A$ sums up to at most $1-\omega$. Together they imply the substochasticity of $A$. The modulus $\omega$ is an important parameter for our finite-sample analysis. In view of Theorem 2.1, we see that large modulus gives smaller (or better) contraction factor of $\tilde{\mathcal{B}}_{c,\rho}(\cdot)$.

**Proposition 2.4.** *For any substochastic matrix $M \in \mathbb{R}^{d\times d}$ with a positive modulus $\beta \in (0,1)$, for any $\theta \in (0,1)$, there exists a weight vector $\mu \in \Delta^d$ satisfying $\mu_i \geq \frac{\beta(1-\theta)}{(1-\theta\beta)d}$ for all $i$ such that $\|M\|_{\mu,p} \leq (1-\beta)^{1-1/p}(1-\theta\beta)^{1/p}$ for any $p \in [1,\infty)$. Furthermore, if $M$ is irreducible [1], then we can choose $\theta = 1$.*

The result of Proposition 2.4 further implies $\|M\|_{\mu,p} \leq 1-\theta\beta$, which is independent of the choice of $p$. This implies that $\tilde{\mathcal{B}}_{c,\rho}(\cdot)$ is a uniform contraction mapping with respect to $\|\cdot\|_{\mu,p}$ for all $p \geq 1$. In general, for different $p$ and $p'$, an operator being a $\|\cdot\|_p$-norm contraction does not imply being a $\|\cdot\|_{p'}$-norm contraction. The reason that we have such a strong uniform contractive result is that the operator $\tilde{\mathcal{B}}_{c,\rho}(\cdot)$ has a linear structure, and involves a substochastic matrix.

Note that Proposition 2.4 introduces the tunable parameter $\theta$. It is clear that large $\theta$ gives better contraction factor of $\tilde{\mathcal{B}}_{c,\rho}(\cdot)$ but worse lower bound on the entries of the weight vector $\mu$. In

---

[1] A non-negative matrix is irreducible if and only if its associated graph is strongly connected [4].

general, when $M$ is not irreducible, we cannot hope to choose a weight vector $\mu \in \Delta^d$ with positive components and obtain $\|M\|_{\mu,p} \leq 1 - \omega$. To see this, consider the example where $M = (1-\omega)[\mathbf{0}, \mathbf{0}, \cdots, \mathbf{1}]$, which is clearly a substochastic matrix with modulus $\omega$, but is not an irreducible matrix. For any weight vector $\mu \in \Delta^d$, we have $\|M\|_{\mu,p} = (1-\omega)\max_{x\in\mathbb{R}^d:\|x\|_{\mu,p}=1}|x_d| = (1-\omega)/\mu_d^{1/p} > 1 - \omega$. However, by choosing $\mu_d$ close to unity, we can get $\|M\|_{\mu,p}$ arbitrarily close to $1 - \omega$. This is analogous to choosing $\theta$ close to one in Proposition 2.4. Since Proposition 2.4 is the major result for proving Theorem 2.1, we provide its proof sketch in Section 4.

## 2.5 Finite-Sample Convergence Guarantees

In light of Theorem 2.1, Algorithm 1 is a Markovian SA algorithm for solving a fixed-point equation $\tilde{\mathcal{B}}_{c,\rho}(Q) = Q$, where the fixed-point operator $\tilde{\mathcal{B}}_{c,\rho}(\cdot)$ is a contraction mapping. Therefore, to establish the finite-sample bounds, we use a Lyapunov drift argument where we choose $W(Q) = \|Q - Q^{\pi,\rho}\|_{\mu,p}^2$ as the Lyapunov function. This leads to a finite-sample bound on $\mathbb{E}[\|Q_k - Q^{\pi,\rho}\|_{\mu,p}^2]$. However, since $\mu$ is unknown, to make the finite-sample bound independent of $\mu$, we use the lower bound on the components of $\mu$ provided in Theorem 2.1, and also tune the parameters $p$ and $\theta$ to obtain a finite-sample bound on $\mathbb{E}[\|Q_k - Q^{\pi,\rho}\|_\infty^2]$. The fact that we have a uniform contraction factor $1 - \theta\omega$ (cf. Theorem 2.1) plays an important role in such tuning process.

To present the results, we need to introduce more notation. For any $\delta > 0$, define $t_\delta(\mathcal{MC}_S)$ as the mixing time of the Markov chain $\{S_k\}$ (induced by $\pi_b$) with precision $\delta$, i.e., $t_\delta(\mathcal{MC}_S) = \min\{k \geq 0 : \max_{s\in\mathcal{S}}\|P^k(s,\cdot) - \kappa_S(\cdot)\|_{\text{TV}} \leq \delta\}$. Under Assumption 2.1, one can easily verify that $t_\delta(\mathcal{MC}_S) \leq L(\log(1/\delta) + 1)$ for some constant $L > 0$, which depends only on $C$ and $\delta$. Let $\tau_{\delta,n} = t_\delta(\mathcal{MC}_S) + n + 1$. The parameters $c_1, c_2$ and $c_3$ used in stating the following theorem are numerical constants, and will be explicitly given in the Appendix. For ease of exposition, we here only present the finite-sample bound for using constant stepsize.

**Theorem 2.2.** *Consider $\{Q_k\}$ of Algorithm 1. Suppose that: (1) Assumptions 2.1 is satisfied, (2) $c(s,a) \leq \rho(s,a)$ for all $(s,a)$ and $D_{\rho,\max} < 1/\gamma$, and (3) the constant stepsize $\alpha$ is chosen such that $\alpha\tau_{\alpha,n} \leq \frac{c_1\omega}{\log(2|\mathcal{S}||\mathcal{A}|/\omega)f(\gamma c_{\max})^2(\gamma\rho_{\max}+1)^2}$. Then we have for all $k \geq \tau_{\alpha,n}$:*

$$\mathbb{E}[\|Q_k - Q^{\pi,\rho}\|_\infty^2] \leq \zeta_1\left(1 - \frac{\omega\alpha}{2}\right)^{k-\tau_{\alpha,n}} + \zeta_2\frac{f(\gamma c_{\max})^2(\gamma\rho_{\max}+1)^2\log(2|\mathcal{S}||\mathcal{A}|/\omega)}{\omega}\alpha\tau_{\alpha,n}, \quad (3)$$

*where $\zeta_1 = c_2(\|Q_0 - Q^{\pi,\rho}\|_\infty + \|Q_0\|_\infty + 1)^2$, and $\zeta_2 = c_3(3\|Q^{\pi,\rho}\|_\infty + 1)^2$.*

Theorem 2.2 enables one to design a wide class of off-policy TD variants with provable finite-sample guarantees by choosing appropriate generalized importance sampling ratios $c(\cdot,\cdot)$ and $\rho(\cdot,\cdot)$. The first term on the RHS of Eq. (3) is usually called the bias in SA literature [6], and it goes to zero at a geometric rate. The second term on the RHS of Eq. (3) stands for the variance in the iterates, and it is a constant proportional to $\alpha\tau_{\alpha,n}$. To see more explicitly the bias-variance trade-off, we derive the sample complexity of Algorithm 1 in the following.

**Corollary 2.2.1.** *For an accuracy $\epsilon > 0$, to obtain $\mathbb{E}[\|Q_k - Q^{\pi,\rho}\|_\infty] \leq \epsilon$, the sample complexity is*

$$\underbrace{\mathcal{O}\left(\frac{\log^2(1/\epsilon)}{\epsilon^2}\right)}_{T_1} \underbrace{\tilde{\mathcal{O}}\left(\frac{1}{\omega^2}\right)}_{T_2} \underbrace{\tilde{\mathcal{O}}\left(\frac{nf(\gamma c_{\max})^2(\gamma\rho_{\max}+1)^2}{(1-\gamma D_{\rho,\max})^2}\right)}_{T_3}. \quad (4)$$

In Corollary 2.2.1, the $\tilde{\mathcal{O}}(\epsilon^{-2})$ dependence on the accuracy is the same as $n$-step TD-learning in the on-policy setting [10], and is in general not improvable. The term $T_2$ can be equivalently written as $\tilde{O}(1/(1 - \text{Contraction factor})^2)$, hence capturing the impact from the contraction factor. This agrees with our intuition that smaller contraction factor leads to better sample complexity. The term $T_3$ arises because of the variance term on the RHS of Eq. (3). The linear dependence on $n$ is due to using $n$-step bootstrapping. By optimizing the sample complexity in terms of $n$, we have $n_{\text{optimal}} \sim 1/\log(1/(\gamma D_{c,\min}))$. This is analogous to the optimal $n$ in the on-policy setting, which is $1/\log(1/\gamma)$ [10]. The additional $D_{c,\min}$ factor arises because of using off-policy learning. The rest of parameters in $T_3$ are determined by the choice of the generalized importance sampling ratios $c(\cdot,\cdot)$ and $\rho(\cdot,\cdot)$. It is clear that smaller $c_{\max}$ and $\rho_{\max}$ lead to smaller variance. As we will see later, this is the reason for the variance reduction of various off-policy TD-learning algorithms in the literature. In light of the previous analysis, the bias-variance trade-off in general off-policy multi-step TD-learning algorithm 1 is intuitively of the form $\tilde{\mathcal{O}}\left(\frac{\text{Variance}}{(1-\text{Contraction factor})^2}\right)$.

# 3 Application to Various Off-Policy TD-Learning Algorithms

In this section, we apply Theorem 2.2 to various off-policy $n$-step TD-learning algorithms in the literature. We begin by introducing some notation. Let $\pi_{\max}$ ($\pi_{\min}$) and $\pi_{b,\max}$ ($\pi_{b,\min}$) be the maximal (minimal) entry of the target policy $\pi$ and the behavior policy $\pi_b$ respectively. Let $r_{\max} = \max_{s,a}(\pi(a|s)/\pi_b(a|s))$ ($r_{\min} = \min_{s,a}(\pi(a|s)/\pi_b(a|s))$) be the maximum (minimum) ratio between $\pi$ and $\pi_b$. We will overload the notation of $\zeta_1$ and $\zeta_2$ from Theorem 2.2. Note that $Q^{\pi,\rho} = Q^\pi$ in $Q^\pi(\lambda)$, TB$(\lambda)$, and Retrace$(\lambda)$, but $Q^{\pi,\rho} \neq Q^\pi$ in $Q$-trace.

## 3.1 Finite-Sample Analysis of Vanilla IS

We first present the sample complexity bound of the Vanilla IS algorithm, where $c(s,a) = \rho(s,a) = \pi(a|s)/\pi_b(a|s)$ for all $(s,a)$.

**Theorem 3.1.** *Consider Algorithm 1 with Vanilla IS update, where we note that $c_{\max} = \rho_{\max} = r_{\max}$, $D_c = D_\rho = I$, and $\omega = \mathcal{K}_{SA,\min}(1 - \gamma^n)$. Suppose that Assumption 2.1 is satisfied. Then, to achieve $\epsilon$-accuracy, the sample complexity is $O\left(\frac{\log^2(1/\epsilon)}{\epsilon^2}\right) \tilde{O}\left(\frac{1}{\omega^2}\right) \tilde{O}\left(\frac{n((\gamma r_{\max})^n + 1)^2}{(1-\gamma)^2}\right)$.*

In the special case where $\pi = \pi_b$ (i.e., on-policy $n$-step TD), the sample complexity bound reduces to $\tilde{O}\left(\frac{n \log^2(1/\epsilon)}{\epsilon^2 \mathcal{K}_{SA,\min}^2 (1-\gamma^n)^2(1-\gamma)^2}\right)$, which is comparable to the results in [10]. See Appendix C.2 for more details. In the off-policy setting, note that the factor $((\gamma r_{\max})^n + 1)^2$ appears in the sample complexity. When $\gamma r_{\max} > 1$ (which can usually happen), the sample complexity bound involves an exponential factor $(\gamma r_{\max})^n$. The reason is that the product of importance sampling ratios $c(\cdot, \cdot)$ are not at all controlled by any means in Vanilla IS. Therefore, the variance can be very large. On the other hand, since the importance sampling ratios are not modified, Vanilla IS effectively uses the full $n$-step return. As a result, the parameter $\omega = \mathcal{K}_{SA,\min}(1 - \gamma^n)$ within Vanilla IS is the largest (best) among all the algorithms we study.

### 3.1.1 Finite-Sample Analysis of $Q^\pi(\lambda)$

In this section, we present the sample complexity of the $Q^\pi(\lambda)$ algorithm, where $c(s,a) = \lambda$ and $\rho(s,a) = \pi(a|s)/\pi_b(a|s)$ for all $(s,a)$.

**Theorem 3.2.** *Consider Algorithm 1 with $Q^\pi(\lambda)$ update, where we note that $c_{\max} = \lambda$, $\rho_{\max} = r_{\max}$, $D_c = \lambda I$, $D_\rho = I$, and $w = \mathcal{K}_{SA,\min}f(\gamma\lambda)(1 - \gamma)$. Suppose that Assumption 2.1 is satisfied, and $\lambda \leq r_{\min}$. Then, to achieve $\epsilon$-accuracy, the sample complexity is $O\left(\frac{\log^2(1/\epsilon)}{\epsilon^2}\right) \tilde{O}\left(\frac{1}{\omega^2}\right) \tilde{O}\left(\frac{nf(\gamma\lambda)^2(\gamma r_{\max}+1)^2}{(1-\gamma)^2}\right)$.*

To see how $Q^\pi(\lambda)$ overcomes the high variance issue in Vanilla IS, observe that since $\gamma\lambda \leq \gamma r_{\min} \leq \gamma < 1$, we have $f^2(\gamma\lambda) \leq 1/(1-\gamma\lambda)^2$. Therefore, by replacing $c(s,a) = \pi(a|s)/\pi_b(a|s)$ in Vanilla IS with a properly chosen constant $\lambda$, $Q^\pi(\lambda)$ algorithm successfully avoids an exponential large factor in the sample complexity. However, choosing a small $\lambda$ to control the variance has a side effect on the contraction factor. Intuitively, when $\lambda$ is small, $Q^\pi(\lambda)$ does not effectively use the $n$-step return. Hence the parameter $\omega$ in $Q^\pi(\lambda)$ is less (worse) than the one in Vanilla IS.

### 3.1.2 Finite-Sample Analysis of TB$(\lambda)$

In this section, we present the sample complexity of the TB$(\lambda)$ algorithm, where $c(s,a) = \lambda\pi(a|s)$ and $\rho(s,a) = \pi(a|s)/\pi_b(a|s)$ for all $(s,a)$.

**Theorem 3.3.** *Consider Algorithm 1 with TB$(\lambda)$ update, where we note that $c_{\max} = \lambda\pi_{\max}$, $\rho_{\max} = r_{\max}$, $D_c(s,a) = \lambda \sum_a \pi_b(a|s)\pi(a|s)$, $D_\rho(s,a) = 1$, and $\omega = \mathcal{K}_{SA,\min}f(\gamma D_{c,\min})(1-\gamma)$. Suppose that Assumption 2.1 is satisfied, and $\lambda \leq 1/\pi_{b,\max}$. Then, to achieve $\epsilon$-accuracy, the sample complexity is $O\left(\frac{\log^2(1/\epsilon)}{\epsilon^2}\right) \tilde{O}\left(\frac{1}{\omega^2}\right) \tilde{O}\left(\frac{nf(\gamma\lambda\pi_{\max})^2(\gamma r_{\max}+1)^2}{(1-\gamma)^2}\right)$.*

Suppose we further choose $\lambda < 1/(\gamma\pi_{\max})$, the TB$(\lambda)$ algorithm also overcomes the high variance issue in Vanilla IS because $f(\gamma\lambda\pi_{\max}) \leq 1/(1 - \gamma\lambda\pi_{\max})$, which does not involve any exponential large factor. When compared to $Q^\pi(\lambda)$, an advantage of TB$(\lambda)$ is that the constraint on $\lambda$ is much relaxed. However, the same side effect on the contraction factor is also present here. To see this, since $D_{c,\min} = \lambda \min_{s,a} \sum_a \pi_b(a|s)\pi(a|s) \leq 1$, the TB$(\lambda)$ algorithm does not effectively use the $n$-step return, hence the parameter $\omega$ in TB$(\lambda)$ is less (worse) than the one in Vanilla IS.

### 3.1.3 Finite-Sample Analysis of Retrace($\lambda$)

In this section, we present the sample complexity of the Retrace($\lambda$) algorithm, where $c(s, a) = \lambda \min(1, \pi(a|s)/\pi_b(a|s))$ and $\rho(s, a) = \pi(a|s)/\pi_b(a|s)$ for all $(s, a)$.

**Theorem 3.4.** *Consider Algorithm 1 with Retrace($\lambda$) update, where we note that $c_{\max} = \lambda$, $\rho_{\max} = r_{\max}$, $D_c(s, a) = \lambda \sum_a \min(\pi_b(a|s), \pi(a|s))$, $D_\rho(s, a) = 1$, and $\omega = \mathcal{K}_{SA,\min} f(\gamma D_{c,\min})(1 - \gamma)$. Suppose that Assumption 2.1 is satisfied, and $\lambda \leq 1$. Then, to achieve $\epsilon$-accuracy, the sample complexity is $O\left(\frac{\log^2(1/\epsilon)}{\epsilon^2}\right) \tilde{\mathcal{O}}\left(\frac{1}{\omega^2}\right) \tilde{\mathcal{O}}\left(\frac{n f(\gamma\lambda)^2(\gamma r_{\max}+1)^2}{(1-\gamma)^2}\right)$.*

The Retrace($\lambda$) algorithm overcomes the high variance issue in Vanilla IS by truncating the importance sampling ratio at 1, which prevents an exponential large factor in the variance term. In addition, it does not require choosing $\lambda$ to be extremely small as required in $Q^\pi(\lambda)$. As for the compromise in the contraction factor, note that $\min(1, \pi(a|s)/\pi_b(a|s)) \geq \pi(a|s)$, which implies that $D_c(s, a)$ (and hence $D_{c,\min}$) is larger in the Retrace($\lambda$) algorithm than the TB($\lambda$) algorithm. As a result, Retrace($\lambda$) does not truncate the $n$-step return as heavy as TB($\lambda$), and hence the parameter $\omega$ is larger (better) in Retrace($\lambda$) than in TB($\lambda$).

### 3.1.4 Finite-Sample Analysis of $Q$-Trace

Lastly, we present the sample complexity of the $Q$-trace algorithm, where $c(s, a) = \min(\bar{c}, \pi(a|s)/\pi_b(a|s))$ and $\rho(s, a) = \min(\bar{\rho}, \pi(a|s)/\pi_b(a|s))$ for all $(s, a)$.

**Theorem 3.5.** *Consider Algorithm 1 with $Q$-trace update, where we note that $c_{\max} = \bar{c}$, $\rho_{\max} = \bar{\rho}$, $D_c(s, a) = \sum_a \min(\bar{c}\pi_b(a|s), \pi(a|s))$, $D_\rho(s, a) = \sum_a \min(\bar{\rho}\pi_b(a|s), \pi(a|s))$, and $\omega = \mathcal{K}_{SA,\min} f(\gamma D_{c,\min})(1 - \gamma D_{\rho,\max})$. Suppose that Assumption 2.1 is satisfied, and $\bar{c} \leq \bar{\rho}$. Then, to achieve $\epsilon$-accuracy, the sample complexity is $O\left(\frac{\log^2(1/\epsilon)}{\epsilon^2}\right) \tilde{\mathcal{O}}\left(\frac{1}{\omega^2}\right) \tilde{\mathcal{O}}\left(\frac{n f(\gamma\bar{c})^2(\gamma\bar{\rho}+1)^2}{(1-\gamma D_{\rho,\max})^2}\right)$.*

To avoid an exponential large variance, in view of the term $f(\gamma\bar{c})$ in our bound, we need to choose $\bar{c} \leq 1/\gamma$. The major difference between $Q$-trace and Retrace($\lambda$) is that the importance sampling ratio $\rho(\cdot, \cdot)$ inside the temporal difference (line 4 of Algorithm 1) also involves a truncation. As shown in Section 2.3, due to introducing the truncation level $\bar{\rho}$, the algorithm converges to a biased limit $Q^{\pi,\rho}$ instead of $Q^\pi$. Such truncation bias can be controlled using Proposition 2.1. These observations agree with the results [19], where the finite-sample bounds of $Q$-trace were first established.

Compared to [19], we have an improved sample complexity. Specifically, the result in [19] implies a sample complexity of $\tilde{\mathcal{O}}(\frac{\log^2(1/\epsilon) n f(\gamma\bar{c})^2(\gamma\bar{\rho}+1)^2}{\epsilon^2 \omega^3 (1-\gamma D_{\rho,\max})^2})$, which has an additional factor of $\omega^{-1}$. Since $\omega^{-1} \propto \mathcal{K}_{SA,\min}^{-1} \geq |\mathcal{S}||\mathcal{A}|$, our result improves the dependency on the size of the state-action space by a factor of at least $|\mathcal{S}||\mathcal{A}|$ compared to [19]. Similarly, since the $V$-trace algorithm [14] is an analog of the $Q$-trace algorithm, we can also improve the sample complexity for $V$-trace in [10].

In addition to analyzing existing algorithms, observe that our results, especially Theorem 2.2, provide sufficient conditions under which Algorithm 1 has provable finite-sample guarantees, and hence enable us to design new algorithms. As an example, in light of the Retrace($\lambda$) algorithm and the $Q$-trace algorithm, one can take advantage of both algorithms to let $c(s, a) = \lambda_c \min(\bar{c}, \pi(a|s)/\pi_b(a|s))$ and $\rho(s, a) = \lambda_\rho \min(\bar{\rho}, \pi(a|s)/\pi_b(a|s))$, where $\lambda_c, \lambda_\rho, \bar{c}$, and $\bar{\rho}$ are tunable parameters. As long as $\lambda_c \bar{c} \leq \lambda_\rho \bar{\rho} < 1/\gamma$, Theorem 2.2 is applicable and hence finite-sample convergence is guaranteed. To avoid an exponentially large variance, we choose $\lambda_c \bar{c} \leq 1/\gamma$ so that there are no exponentially large terms in the term $T_3$ of sample complexity bound. After that, we can tune the rest of the parameters to further optimize the performance of the algorithm.

**Sample Complexity Comparison.** Now that we have derived the sample complexity bounds of various off-policy $n$-step TD-learning algorithms, we summarize them in the following table. For ease of exposition, we omit the common factor $\log^2(1/\epsilon)/(\epsilon^2 \mathcal{K}_{SA,\min}^2)$ when presenting the sample complexity, and use $a \wedge b$ ($a \vee b$) to denote the minimum (maximum) of two real numbers $a$ and $b$.

In view of Table 1, when $r_{\max} < 1/\gamma$, which indicates that the target policy $\pi$ and the behavior policy $\pi_b$ are relatively close to each other, Vanilla IS has the best performance since it has the best contraction factor, and the cumulative product of the generalized importance sampling ratios does not result in exponentially large variance. When $r_{\max} > 1/\gamma$, then Vanilla IS can potentially have exponentially large variance, while other four algorithms do not. In this case, among $Q^\pi(\lambda)$, TB($\lambda$), and Retrace($\lambda$), $Q^\pi(\lambda)$ has the best sample complexity bound. However, we need to point out that

Table 1: Summary of the Sample Complexity Bounds

| Algorithm | $c(s,a)$ | $\rho(s,a)$ | Requirements | Sample Complexity |
|---|---|---|---|---|
| Vanilla IS | $\frac{\pi(a\|s)}{\pi_b(a\|s)}$ | $\frac{\pi(a\|s)}{\pi_b(a\|s)}$ | None | $\tilde{\mathcal{O}}\left(\frac{(\gamma r_{\max})^n+1)^2}{(1-\gamma^n)^2(1-\gamma)^2}\right)$ |
| $Q^\pi(\lambda)$ | $\lambda$ | $\frac{\pi(a\|s)}{\pi_b(a\|s)}$ | $\lambda \leq r_{\min}$ | $\tilde{\mathcal{O}}\left(\frac{(\gamma r_{\max}+1)^2}{(1-\gamma)^4}\right)$ |
| TB$(\lambda)$ | $\lambda\pi(a\|s)$ | $\frac{\pi(a\|s)}{\pi_b(a\|s)}$ | $\lambda < \frac{1}{(\pi_{b,\max}\vee\gamma\pi_{\max})}$ | $\tilde{\mathcal{O}}\left(\frac{f(\gamma\lambda\pi_{\max})^2(\gamma r_{\max}+1)^2}{f(\gamma D_{c,\min})^2(1-\gamma)^4}\right)$ |
| Retrace$(\lambda)$ | $\lambda[1\wedge\frac{\pi(a\|s)}{\pi_b(a\|s)}]$ | $\frac{\pi(a\|s)}{\pi_b(a\|s)}$ | $\lambda \leq 1$ | $\tilde{\mathcal{O}}\left(\frac{f(\gamma\lambda)^2(\gamma r_{\max}+1)^2}{f(\gamma D_{c,\min})^2(1-\gamma)^4}\right)$ |
| $Q$-trace | $\bar{c}\wedge\frac{\pi(a\|s)}{\pi_b(a\|s)}$ | $\bar{\rho}\wedge\frac{\pi(a\|s)}{\pi_b(a\|s)}$ | $\bar{c}\leq\bar{\rho},\ \bar{c}<\frac{1}{\gamma}$ | $\tilde{\mathcal{O}}\left(\frac{f(\gamma\bar{c})^2(\gamma\bar{\rho}+1)^2}{f(\gamma D_{c,\min})^2(1-\gamma D_{\rho,\max})^4}\right)$ |

the requirement $\lambda \leq r_{\min}$ for $Q^\pi(\lambda)$ is most restrictive, and the algorithm can easily diverge when this requirement is not satisfied, as evidenced by the numerical experiments presented in [22]. As for the $Q$-trace algorithm, although rigorously speaking it is not directly comparable with the other algorithms as it converges to a biased limit point, it is clear that using truncated importance sampling ratio for $\rho(\cdot,\cdot)$ can further reduce the sample complexity.

We want to mention that our comparison is based on the upper bounds we derived for the sample complexity, which may not be tight. To complete the story, one should also derive lower bounds on the sample complexity, which is an interesting future direction. Nevertheless, our comparison provides insight into the behavior of off-policy $n$-step TD-learning algorithms,

## 4 Proof sketch of Proposition 2.4

The idea is to construct a stochastic matrix $M''$ such that: (1) $M''$ dominates $M$ in the sense that $M''_{ij} \geq M_{ij}$ for all $i,j$, and (2) the Markov chain associated with $M''$ is irreducible, hence admits a unique stationary distribution $\mu > 0$. Using $\mu$ as weights and we have the desired result. The detailed analysis is presented in Appendix A.5. We here present the construction of the stochastic matrix $M''$.

First of all, consider the special case where $M$ itself is irreducible. Then we first scale up $M$ by a factor of $1/(1-\omega)$ to obtain $M' = \frac{M}{1-\omega}$, which is clearly a substochastic matrix, with modulus zero. Hence there exists a stochastic matrix $M''$ that dominates $M'$ (and also $M$). Moreover, since $M''$ is also irreducible, its associated Markov chain has a unique stationary distribution $\mu$. This is equivalent to choosing $\theta = 1$ in Proposition 2.4. In fact, the matrix $M$ being irreducible is only a sufficient condition to choose $\theta = 1$. What we need is the existence of a strictly positive stationary distribution of the stochastic matrix $M''$, which is guaranteed when $M''$ does not have transient states.

Now consider the general case where $M$ is not necessarily irreducible. We construct the intermediate matrix $M'$ by performing a convex combination of the matrix $\frac{M}{1-\omega}$ and the uniform stochastic matrix $\frac{E}{d}$, where $E$ is the all one matrix, with weight $\frac{1-\omega}{1-\theta\omega}$. Specifically, for any $\theta \in (0,1)$, we define $M' = \left(\frac{1-\omega}{1-\theta\omega}\right)\frac{M}{1-\omega} + \left(1 - \frac{1-\omega}{1-\theta\omega}\right)\frac{E}{d}$. Note that $M'$ is a non-negative matrix. In addition, since $M'\mathbf{1} \leq \frac{1-\omega}{1-\theta\omega}\mathbf{1} + \left(1 - \frac{1-\omega}{1-\theta\omega}\right)\mathbf{1} = \mathbf{1}$, where $\mathbf{1}$ is the all one vector, the matrix $M'$ is a substochastic matrix with modulus zero, and is also irreducible because all its entries are strictly positive. Therefore, there exists a stochastic matrix $M''$ such that $M'' \geq M'$. In addition, since $M''$ also has strictly positive entries, the Markov chain associated with $M''$ is irreducible, hence admits a unique stationary distribution $\mu \in \Delta^d$. By our construction, we can show a lower bound on the components of the stationary distribution $\mu$.

## 5 Conclusion

In this work, we establish finite-sample guarantees of general $n$-step off-policy TD-learning algorithms. The key in our approach is to identify a generalized Bellman operator and establish its contraction property with respect to a weighted $\ell_p$-norm for each $p \in [1, \infty)$, with a uniform contraction factor. Our results are used to derive finite-sample guarantees of variants of $n$-step off-policy TD-learning algorithms in the literature. Specifically, for $Q^\pi(\lambda)$, TB$(\lambda)$, and Retrace$(\lambda)$, we provide the first-known results, and for $Q$-trace, we improve the result in [19]. The finite-sample bounds we establish also provide insights about the trade-offs between the bias and the variance.

## Acknowledgements

This work was partially supported by ONR Grant N00014-19-1-2566, NSF Grants 1910112, 2019844, NSF Grant CCF-1740776, and an award from Raytheon Technologies. Maguluri acknowledges seed funding from Georgia Institute of Technology.

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
