# Appendices

## A Technical Details in Section 2

### A.1 Proof of Proposition 2.1

For any $Q_1, Q_2 \in \mathbb{R}^{|\mathcal{S}||\mathcal{A}|}$, and state-action pairs $(s, a)$, using the definition of $\mathcal{H}_\rho(\cdot)$ and we have

$$|[\mathcal{H}_\rho(Q_1)](s, a) - [\mathcal{H}_\rho(Q_2)](s, a)|$$

$$= \gamma \left| \sum_{s' \in \mathcal{A}} P_a(s, s') \sum_{a' \in \mathcal{A}} \pi_b(a'|s')\rho(s', a')(Q_1(s', a') - Q_2(s', a')) \right|$$

$$\leq \gamma \sum_{s' \in \mathcal{A}} P_a(s, s') \sum_{a' \in \mathcal{A}} \pi_b(a'|s')\rho(s', a')|Q_1(s', a') - Q_2(s', a')|$$

$$\leq \gamma \|Q_1 - Q_2\|_\infty \sum_{s' \in \mathcal{A}} P_a(s, s') \sum_{a' \in \mathcal{A}} \pi_b(a'|s')\rho(s', a')$$

$$\leq \gamma \sum_{s' \in \mathcal{A}} P_a(s, s') D_{\rho,\max} \|Q_1 - Q_2\|_\infty$$

$$= \gamma D_{\rho,\max} \|Q_1 - Q_2\|_\infty.$$

It follows that $\|\mathcal{H}_\rho(Q_1) - \mathcal{H}_\rho(Q_2)\|_\infty \leq \gamma D_{\rho,\max} \|Q_1 - Q_2\|_\infty$. Since $D_{\rho,\max} < 1/\gamma$, the operator $\mathcal{H}_\rho(\cdot)$ is a contraction mapping with respect to $\| \cdot \|_\infty$, with contraction factor $\gamma D_{\rho,\max}$.

(1) We now derive the upper bound on $\|Q^\pi - Q^{\pi,\rho}\|_\infty$. Since $Q^\pi = \mathcal{H}_\pi(Q^\pi)$ and $Q^{\pi,\rho} = \mathcal{H}_\rho(Q^{\pi,\rho})$, we have

$$|Q^\pi(s, a) - Q^{\pi,\rho}(s, a)|$$
$$= |[\mathcal{H}_\pi(Q^\pi)](s, a) - [\mathcal{H}_\rho(Q^{\pi,\rho})](s, a)|$$
$$= |[\mathcal{H}_\pi(Q^\pi)](s, a) - [\mathcal{H}_\rho(Q^\pi)](s, a) + [\mathcal{H}_\rho(Q^\pi)](s, a) - [\mathcal{H}_\rho(Q^{\pi,\rho})](s, a)|$$
$$\leq |[\mathcal{H}_\pi(Q^\pi)](s, a) - [\mathcal{H}_\rho(Q^\pi)](s, a)| + |[\mathcal{H}_\rho(Q^\pi)](s, a) - [\mathcal{H}_\rho(Q^{\pi,\rho})](s, a)|$$
$$= \gamma \left| \sum_{s' \in \mathcal{S}} P_a(s, s') \sum_{a' \in \mathcal{A}} (\pi(a'|s') - \pi_b(a'|s')\rho(s', a')) Q^\pi(s', a') \right| + \gamma D_{\rho,\max} \|Q^\pi - Q^{\pi,\rho}\|_\infty$$
$$\leq \frac{\gamma}{1-\gamma} \sum_{s' \in \mathcal{S}} P_a(s, s') \sum_{a' \in \mathcal{A}} |\pi(a'|s') - \pi_b(a'|s')\rho(s', a')| + \gamma D_{\rho,\max} \|Q^\pi - Q^{\pi,\rho}\|_\infty \qquad (*)$$
$$\leq \frac{\gamma}{1-\gamma} \max_{s \in \mathcal{S}} \sum_{a \in \mathcal{A}} |\pi(a|s) - \pi_b(a|s)\rho(s, a)| + \gamma D_{\rho,\max} \|Q^\pi - Q^{\pi,\rho}\|_\infty,$$

where in Eq. $(*)$ we used the inequality $|Q^\pi(s, a)| \leq \sum_{k=0}^\infty \gamma^k = \frac{1}{1-\gamma}$ for all $(s, a)$. Therefore, we have

$$\|Q^\pi - Q^{\pi_\rho}\|_\infty \leq \frac{\gamma}{1-\gamma} \max_{s \in \mathcal{S}} \sum_{a \in \mathcal{A}} |\pi(a|s) - \pi_b(a|s)\rho(s, a)| + \gamma D_{\rho,\max} \|Q^\pi - Q^{\pi,\rho}\|_\infty.$$

Rearranging terms and we obtain the desired result.

(2) To prove the upper bound on $\|Q^{\pi,\rho}\|_\infty$, we begin with the fixed-point equation

$$Q^{\pi,\rho} = \mathcal{H}_\rho(Q^{\pi,\rho}) = R + \gamma P_{\pi_\rho} D_\rho Q^{\pi,\rho}, \tag{5}$$

where we recall the definition of $D_\rho$ and $\pi_\rho$ in Section 2. Eq. (5) is equivalent to $Q^{\pi,\rho} = (I - \gamma P_{\pi_\rho} D_\rho)^{-1} R$. Therefore, we have

$$\|Q^{\pi,\rho}\|_\infty = \|(I - \gamma P_{\pi_\rho} D_\rho)^{-1} R\|_\infty \leq \|(I - \gamma P_{\pi_\rho} D_\rho)^{-1}\|_\infty \|R\|_\infty \leq \frac{1}{1 - \gamma D_{\rho,\max}}.$$

## A.2 On the Fixed-Point of the Operator $\mathcal{H}_\rho(\cdot)$

Suppose that for the state-action pair $(s_0, a_0)$, we have $\rho(s_0, a_0) \neq \pi(a_0|s_0)/\pi_b(a_0|s_0)$. Let an MDP be that the transition probability matrix is an identity matrix for each action, and the reward is zero for all state-action pairs except at $(s_0, a_0)$, where it is equal to 1.

In this case, it is clear that for any policy $\pi$, we have $Q^\pi(s, a) = 0$ for all $(s, a) \neq (s_0, a_0)$, and $Q^\pi(s_0, a_0) = \frac{1}{1-\gamma}$. Suppose that $Q^\pi = Q^{\pi,\rho}$. then we have

$$
\begin{aligned}
0 &= Q^\pi(s_0, a_0) - Q^{\pi,\rho}(s_0, a_0) \\
&= \gamma \sum_{s' \in \mathcal{S}} P_{a_0}(s_0, s') \sum_{a' \in \mathcal{A}} \left( \pi(a'|s') - \pi_b(a'|s')\rho(s', a') \right) Q^\pi(s', a') \\
&= \left( \pi(a_0|s_0) - \pi_b(a_0|s_0)\rho(s_0, a_0) \right) Q^\pi(s_0, a_0) \\
&= \frac{1}{1-\gamma} \left( \pi(a_0|s_0) - \pi_b(a_0|s_0)\rho(s_0, a_0) \right).
\end{aligned}
$$

This contradicts to the fact that $\rho(s_0, a_0) \neq \pi(a_0|s_0)/\pi_b(a_0|s_0)$. Therefore, we have $Q^\pi \neq Q^{\pi,\rho}$.

## A.3 Proof of Proposition 2.2

Recall the definition of $\tilde{\mathcal{B}}_{c,\rho}(\cdot)$ in Eq. (2):

$$
\tilde{\mathcal{B}}_{c,\rho}(Q) = \mathcal{K}_{SA}(\mathcal{B}_{c,\rho}(Q) - Q) + Q = \mathcal{K}_{SA}\mathcal{T}_c(\mathcal{H}_\rho(Q) - Q) + Q.
$$

We first explicitly compute the operators $\mathcal{T}_c(\cdot)$ and $\mathcal{H}_\rho(\cdot)$. For the operator $\mathcal{H}_\rho(\cdot)$, we have from its definition that

$$
\begin{aligned}
[\mathcal{H}_\rho(Q)](s, a) &= \mathcal{R}(s, a) + \gamma \mathbb{E}_{\pi_b}[\rho(S_{k+1}, A_{k+1})Q(S_{k+1}, A_{k+1}) \mid S_k = s, A_k = a] \\
&= \mathcal{R}(s, a) + \gamma \sum_{s'} P_a(s, s') \sum_{a'} \pi_b(a'|s')\rho(s', a')Q(s', a') \\
&= \mathcal{R}(s, a) + \gamma \sum_{s'} P_a(s, s') \sum_{a'} \frac{\pi_b(a'|s')\rho(s', a')}{D_\rho(s', a')} D_\rho(s', a')Q(s', a') \\
&= \mathcal{R}(s, a) + \gamma \sum_{s', a'} P_a(s, s')\pi_\rho(a'|s')D_\rho(s', a')Q(s', a') \\
&= [R + P_{\pi_\rho}D_\rho Q](s, a).
\end{aligned}
$$

Note that $P_{\pi_\rho} \in \mathbb{R}^{|\mathcal{S}||\mathcal{A}| \times |\mathcal{S}||\mathcal{A}|}$ here is the transition probability matrix of the Markov chain $\{(S_k, A_k)\}$ under $\pi_\rho$, i.e., $P_{\pi_\rho}((s, a), (s', a')) = P_a(s, s')\pi_\rho(a'|s')$ for any $(s, a)$ and $(s', a')$. Hence we have

$$
\mathcal{H}_\rho(Q) = R + P_{\pi_\rho}D_\rho Q.
$$

As for the operator $\mathcal{T}_c(\cdot)$, similarly using the Markov property and the tower property of conditional expectation, we have $\mathcal{T}_c(Q) = \sum_{i=0}^{n-1}(\gamma P_{\pi_c}D_c)^i Q$. It follows that

$$
\begin{aligned}
\tilde{\mathcal{B}}_{c,\rho}(Q) &= \mathcal{K}_{SA}\mathcal{T}_c(\mathcal{H}_\rho(Q) - Q) + Q \\
&= \mathcal{K}_{SA}\sum_{i=0}^{n-1}(\gamma P_{\pi_c}D_c)^i(R + \gamma P_{\pi_\rho}D_\rho Q - Q) + Q \\
&= \underbrace{\left[ I - \mathcal{K}_{SA}\sum_{i=0}^{n-1}(\gamma P_{\pi_c}D_c)^i(I - \gamma P_{\pi_\rho}D_\rho) \right]}_{A} Q + \underbrace{\mathcal{K}_{SA}\sum_{i=0}^{n-1}(\gamma P_{\pi_c}D_c)^i R}_{b}.
\end{aligned}
$$

## A.4 Proof of Proposition 2.3

Consider the matrix $A$ given in Proposition 2.2. To show that $A$ is a substochastic matrix with a positive modulus, we first show that $A$ is non-negative. Observe that

$$
A = I - \mathcal{K}_{SA}\sum_{i=0}^{n-1}(\gamma P_{\pi_c}D_c)^i + \mathcal{K}_{SA}\sum_{i=0}^{n-1}(\gamma P_{\pi_c}D_c)^i \gamma P_{\pi_\rho}D_\rho
$$

$$= (I - \mathcal{K}_{SA}) - \mathcal{K}_{SA} \sum_{i=1}^{n-1} (\gamma P_{\pi_c} D_c)^i + \mathcal{K}_{SA} \sum_{i=0}^{n-1} (\gamma P_{\pi_c} D_c)^i \gamma P_{\pi_\rho} D_\rho$$

$$= (I - \mathcal{K}_{SA}) - \mathcal{K}_{SA} \sum_{i=0}^{n-2} (\gamma P_{\pi_c} D_c)^{i+1} + \mathcal{K}_{SA} \sum_{i=0}^{n-1} (\gamma P_{\pi_c} D_c)^i \gamma P_{\pi_\rho} D_\rho$$

$$= (I - \mathcal{K}_{SA}) + \mathcal{K}_{SA} \sum_{i=0}^{n-2} (\gamma P_{\pi_c} D_c)^i \gamma (P_{\pi_\rho} D_\rho - P_{\pi_c} D_c) + \mathcal{K}_{SA} (\gamma P_{\pi_c} D_c)^{n-1} \gamma P_{\pi_\rho} D_\rho. \quad (6)$$

It remains to show that the matrix $P_{\pi_\rho} D_\rho - P_{\pi_c} D_c$ has non-negative entries. For any $(s, a)$ and $(s', a')$, since $c(s', a') \leq \rho(s', a')$ for all $(s', a')$, we have

$$[P_{\pi_\rho} D_\rho - P_{\pi_c} D_c]((s, a), (s', a')) = P_a(s, s') \pi_b(a'|s')(\rho(s', a') - c(s', a')) \geq 0.$$

We next show that $A\mathbf{1} \leq (1 - \omega)\mathbf{1}$, where $\mathbf{1} \in \mathbb{R}^d$ is the all one vector. Since $A$ is non-negative and $D_{\rho,\max} < 1/\gamma$ for all $(s, a)$, we have

$$\mathcal{K}_{SA} \sum_{i=0}^{n-1} (\gamma P_{\pi_c} D_c)^i (I - \gamma P_{\pi_\rho} D_\rho)\mathbf{1} \geq \mathcal{K}_{SA} \sum_{i=0}^{n-1} (\gamma P_{\pi_c} D_c)^i (I - \gamma P_{\pi_\rho} D_{\rho,\max})\mathbf{1}$$

$$= (1 - \gamma D_{\rho,\max}) \mathcal{K}_{SA} \sum_{i=0}^{n-1} (\gamma P_{\pi_c} D_c)^i \mathbf{1}$$

$$\geq \mathcal{K}_{SA,\min} \sum_{i=0}^{n-1} (\gamma D_{c,\min})^i (1 - \gamma D_{\rho,\max})\mathbf{1}$$

$$= \mathcal{K}_{SA,\min} f(\gamma D_{c,\min})(1 - \gamma D_{\rho,\max})\mathbf{1}.$$

It follows that

$$A\mathbf{1} = \left[ I - \mathcal{K}_{SA} \sum_{i=0}^{n-1} (\gamma P_{\pi_c} D_c)^i (I - \gamma P_{\pi_\rho} D_\rho) \right] \mathbf{1} \leq [1 - \mathcal{K}_{SA,\min} f(\gamma D_{c,\min})(1 - \gamma D_{\rho,\max})]\mathbf{1}.$$

This implies that $A$ is a substochastic matrix with modulus $\omega = \mathcal{K}_{SA,\min} f(\gamma D_{c,\min})(1 - \gamma D_{\rho,\max})$.

### A.5 Proof of Proposition 2.4

Consider a substochastic matrix $M \in \mathbb{R}^{d \times d}$ with modulus $\beta \in (0, 1)$. For any $\theta \in (0, 1)$, let

$$M' = \frac{M}{1 - \theta\beta} + \frac{\beta(1 - \theta)}{1 - \theta\beta} \frac{E}{d},$$

where $E$ is the all one matrix. It is clear that $M' > 0$. Moreover, since

$$M'\mathbf{1} \leq \frac{1 - \beta}{1 - \theta\beta}\mathbf{1} + \frac{\beta(1 - \theta)}{1 - \theta\beta}\mathbf{1} = \mathbf{1},$$

the matrix $M'$ is a substochastic matrix with modulus 0, there exists a stochastic matrix $M''$ such that $M'' \geq M' > 0$. Since $M''$ has strictly positive entries, the Markov chain associated with the stochastic matrix $M''$ is irreducible and aperiodic, hence admits a unique stationary distribution $\mu \in \Delta^d$. In the special case where $M$ itself is irreducible, we are allowed to choose $\theta = 1$ in the preceding construction process, and the resulting stochastic matrix $M''$ is also guaranteed to be irreducible, and hence has a unique stationary distribution $\mu$. Since $\mu^\top = \mu^\top M''$, we have

$$\mu^\top = \mu^\top M'' \geq \mu^\top M' \geq \mu^\top \frac{\beta(1 - \theta)}{1 - \theta\beta} \frac{E}{d} = \frac{\beta(1 - \theta)}{(1 - \theta\beta)d}\mathbf{1}^\top.$$

This proves the lower bound on the entries of $\mu$.

Now using $\mu$ as the weight vector and we have for any $p \in [1, \infty)$ and $x \in \mathbb{R}^d$:

$$\|Mx\|_{\mu,p}^p = \sum_i \mu_i \left| \sum_j M_{ij} x_j \right|^p$$

$$= \sum_i \mu_i \left( \sum_\ell M_{i\ell} \right)^p \left| \sum_j \frac{M_{ij}}{\sum_\ell M_{i\ell}} x_j \right|^p$$

$$\leq \sum_i \mu_i \left( \sum_\ell M_{i\ell} \right)^{p-1} \sum_j M_{ij} |x_j|^p \qquad \text{(Jensen's inequality)}$$

$$\leq (1-\beta)^{p-1} \sum_i \mu_i \sum_j M_{ij} |x_j|^p$$

$$\leq (1-\beta)^{p-1} (1-\theta\beta) \sum_i \mu_i \sum_j M'_{ij} |x_j|^p \qquad \text{(definition of } M')$$

$$\leq (1-\beta)^{p-1} (1-\theta\beta) \sum_i \mu_i \sum_j M''_{ij} |x_j|^p \qquad \text{(definition of } M'')$$

$$= (1-\beta)^{p-1} (1-\theta\beta) \sum_j |x_j|^p \sum_i \mu_i M''_{ij} \qquad \text{(change of summation order)}$$

$$= (1-\beta)^{p-1} (1-\theta\beta) \sum_j \mu_j |x_j|^p \qquad (\mu^\top M'' = \mu^\top)$$

$$= (1-\beta)^{p-1} (1-\theta\beta) \|x\|_{\mu,p}^p.$$

It follows that $\|Mx\|_{\mu,p} \leq (1-\omega)^{1-1/p}(1-\theta\beta)^{1/p} \|x\|_{\mu,p}$ for any $x \in \mathbb{R}^d$ and $p \in [1,\infty)$. Using the definition of induced matrix norm immediately gives the result.

## A.6 Proof of Theorem 2.2

We first state a more general result in the following, which implies Theorem 2.2.

**Theorem A.1.** *Consider the iterates $\{Q_k\}$ generated by Algorithm 1. Suppose that Assumption 2.1 is satisfied, and $c(s,a) \leq \rho(s,a)$ for all $(s,a)$ and $D_{\rho,\max} < 1/\gamma$. Then for any $\theta \in (0,1)$, there exists a weighted $\ell_p$-norm with weights $\mu \in \Delta^{|\mathcal{S}||\mathcal{A}|}$ satisfying $\mu_{\min} \geq \frac{\omega(1-\theta)}{(1-\theta\omega)|\mathcal{S}||\mathcal{A}|}$ such that the following inequality holds when the constant stepsize $\alpha$ is chosen such that $\alpha\tau_{\alpha,n} \leq \frac{\theta\mu_{\min}^{2/p}\omega}{2052pf(\gamma c_{\max})^2(\gamma\rho_{\max}+1)^2}$:*

$$\mathbb{E}[\|Q_k - Q^{\pi,\rho}\|_{\mu,p}^2] \leq \tilde{\zeta}_1 (1-\theta\omega\alpha)^{k-\tau_{\alpha,n}} + \tilde{\zeta}_2 \frac{pf(\gamma c_{\max})^2(\gamma\rho_{\max}+1)^2}{\mu_{\min}^{2/p}\omega}\alpha\tau_{\alpha,n},$$

*where $\tilde{\zeta}_1 = (\|Q_0 - Q^{\pi,\rho}\|_{\mu,p} + \|Q_0\|_{\mu,p} + 1)^2$, and $\tilde{\zeta}_2 = 228(3\|Q^{\pi,\rho}\|_{\mu,p} + 1)^2$.*

By using the inequality that $\mu_{\min}^{1/p} \|\cdot\|_p \leq \|\cdot\|_{\mu,p}$ (where $\|\cdot\|_p$ is the unweighted $\ell_p$-norm), Theorem A.1 implies the following finite-sample bound on $\mathbb{E}[\|Q_k - Q^{\pi,\rho}\|_p]$.

**Corollary A.1.1.** *Under same assumptions as Theorem 2.1, we have for all $k \geq \tau_{\alpha,n}$:*

$$\mathbb{E}[\|Q_k - Q^{\pi,\rho}\|_p^2] \leq \frac{\tilde{\zeta}_1}{\mu_{\min}^{2/p}} (1-\theta\omega\alpha)^{k-\tau_{\alpha,n}} + \frac{\tilde{\zeta}_2}{\mu_{\min}^{2/p}} \frac{pf(\gamma c_{\max})^2(\gamma\rho_{\max}+1)^2}{\mu_{\min}^{2/p}\omega}\alpha\tau_{\alpha,n},$$

To proceed and prove Theorem 2.2, observe that for any $p \geq 1$ we have

$$\mathbb{E}[\|Q_k - Q^{\pi,\rho}\|_\infty^2] \leq \mathbb{E}[\|Q_k - Q^{\pi,\rho}\|_p^2]$$

$$\leq \frac{\tilde{\zeta}_1}{\mu_{\min}^{2/p}} (1-\theta\omega\alpha)^{k-\tau_{\alpha,n}} + \frac{\tilde{\zeta}_2 pf(\gamma c_{\max})^2(\gamma\rho_{\max}+1)^2}{\mu_{\min}^{4/p}\omega}\alpha\tau_{\alpha,n}.$$

Let $\theta = 1/2$ and $p = 4\log(1/\mu_{\min})$. Then we have

$$\frac{1}{\mu_{\min}^{2/p}} = \mu_{\min}^{-\frac{1}{2\log(1/\mu_{\min})}} = \mu_{\min}^{\frac{1}{2\log(\mu_{\min})}} = \sqrt{e} \leq 2, \quad \text{and}$$

$$\frac{p}{\mu_{\min}^{4/p}} \leq 4e\log(1/\mu_{\min}) \leq 4e\log\left(\frac{2|\mathcal{S}||\mathcal{A}|}{\omega}\right). \qquad \text{(Using the lower bound on } \mu_{\min})$$

It follows that when $\alpha\tau_{\alpha,n} \le \frac{\omega}{32832\log(2|\mathcal{S}||\mathcal{A}|/\omega)f(\gamma c_{\max})^2(\gamma\rho_{\max}+1)^2}$, we have for all $k \ge \tau_{\alpha,n}$:

$$\mathbb{E}[\|Q_k - Q^{\pi,\rho}\|_\infty^2] \le 2\tilde{\zeta}_1\left(1 - \frac{\omega\alpha}{2}\right)^{k-\tau_{\alpha,n}} + 4e\tilde{\zeta}_2\frac{f(\gamma c_{\max})^2(\gamma\rho_{\max}+1)^2\log(2|\mathcal{S}||\mathcal{A}|/\omega)}{\omega}\alpha\tau_{\alpha,n}$$

$$= \zeta_1\left(1 - \frac{\omega\alpha}{2}\right)^{k-\tau_{\alpha,n}} + \zeta_2\frac{f(\gamma c_{\max})^2(\gamma\rho_{\max}+1)^2\log(2|\mathcal{S}||\mathcal{A}|/\omega)}{\omega}\alpha\tau_{\alpha,n},$$

where in the last line we used $2\tilde{\zeta}_1 \le \zeta_1 = 2(\|Q_0 - Q^{\pi,\rho}\|_\infty + \|Q_0\|_\infty + 1)^2$, and $4e\tilde{\zeta}_2 \le \zeta_2 = 912e(3\|Q^{\pi,\rho}\|_\infty + 1)^2$. This proves Theorem 2.2.

### A.6.1 Proof of Theorem A.1

To prove Theorem A.1, we use a Lyapunov drift argument. We next present two approaches for proving Theorem A.1. One is by directly using $W(Q) = \frac{1}{2}\|Q\|_{\mu,p}^2$ as the Lyapunov function. Another one is by applying [10, Theorem 2.1], which studies general stochastic approximation under contraction assumption.

We begin by rewriting Algorithm 1 using simplified notation. Let $Y_k = (S_k, A_k, \cdots, S_{k+n}, A_{k+n})$ for all $k \ge 0$, which is clearly a Markov chain, with finite state-space denoted by $\mathcal{Y}$. Note that under Assumption 2.1 the Markov chain $\{Y_k\}$ has a unique stationary distribution $\kappa_Y \in \Delta^{|\mathcal{Y}|}$. Define an operator $F : \mathbb{R}^{|\mathcal{S}||\mathcal{A}|} \times \mathcal{Y} \mapsto \mathbb{R}^{|\mathcal{S}||\mathcal{A}|}$ by

$$[F(Q,y)](s,a) = [F(Q, s_0, a_0, ..., s_n, a_n)](s,a)$$

$$= \mathbb{I}_{\{(s_0,a_0)=(s,a)\}}\sum_{i=0}^{n-1}\gamma^i\prod_{j=1}^{i}c(s_j,a_j)(\mathcal{R}(s_i,a_i)+\gamma\rho(s_{i+1},a_{i+1})Q(s_{i+1},a_{i+1})-Q(s_i,a_i))+Q(s,a).$$

Then the update equation of Algorithm 1 can be equivalently written by $Q_{k+1} = Q_k + \alpha(F(Q_k, Y_k) - Q_k)$. We next establish in the following proposition the properties of the operators $F(\cdot, \cdot)$ and the Markov chain $\{Y_k\}$, which will be useful in both approaches we present later.

**Proposition A.1.** *The following statements hold.*

*(1) The operator $F(\cdot)$ satisfies for any $Q_1, Q_2$ and $y$:*

    *(a) $\|F(Q_1, y) - F(Q_2, y)\|_{\mu,p} \le \frac{2}{\mu_{\min}^{1/p}}f(\gamma c_{\max})(\gamma\rho_{\max}+1)\|Q_1 - Q_2\|_{\mu,p}$,*

    *(b) $\|F(\mathbf{0}, y)\|_{\mu,p} \le f(\gamma c_{\max})$.*

*(2) For any $k \ge 0$ and $n \ge 0$, we have $\max_{y\in\mathcal{Y}}\|P^{k+n+1}(y,\cdot) - \kappa_Y(\cdot)\|_{TV} \le C\sigma^k$.*

*(3) For any $Q$, we have $\mathbb{E}_{Y\sim\kappa_Y}[F(Q,Y)] = \tilde{\mathcal{B}}_{c,\rho}(Q)$.*

We now present our first approach of proving Theorem A.1, where we directly use $W(Q) = \frac{1}{2}\|Q\|_{\mu,p}^2$ as the Lyapunov function.

**First Approach:** Note that the function $W(Q) = \frac{1}{2}\|Q\|_{\mu,p}^2$ is a $(p-1)$-smooth function with respect to $\|\cdot\|_{\mu,p}$ [3], i.e., $W(Q_2) \le W(Q_1) + \langle\nabla W(Q_1), Q_2 - Q_1\rangle + \frac{p-1}{2}\|Q_1 - Q_2\|_{\mu,p}^2$ for any $Q_1, Q_2 \in \mathbb{R}^{|\mathcal{S}||\mathcal{A}|}$. Therefore, using the update equation of Algorithm 1, we have for any $k \ge 0$:

$$\mathbb{E}[W(Q_{k+1} - Q^{\pi,\rho})] \le \mathbb{E}[W(Q_k - Q^{\pi,\rho})] + \alpha_k\mathbb{E}[\langle\nabla W(Q_k - Q^{\pi,\rho}), Q_{k+1} - Q_k\rangle]$$

$$+ \frac{(p-1)\alpha_k^2}{2}\mathbb{E}[\|Q_{k+1} - Q_k\|_p^2].$$

The rest of the proof is identical to that of [10, Theorem 2.1] (where Proposition A.1 plays an important role), and is omitted. Here, we can directly use $W(Q)$ as a Lyapunov function because it is smooth. In contrast, [9, 10] study the more general settings when it is not smooth. In that case, the Lyapunov function is obtained by using a smoothing technique involving generalized Moreau envelop and infimal convolution to obtain a smooth approximation of $W(Q)$. One can of course, directly apply the result in [9, 10], which we present as a second approach.

**Second Approach:** We next present how to apply [10, Theorem 2.1] to obtain the results. We begin by restating Theorem 2.1 of [10] in the case of weighted $\ell_p$-norm contraction with weights $\{\mu_i\}_{1 \le i \le d}$. Using the notation of [10], we choose the smoothing norm $\|\cdot\|_s$ to be the same norm as the contraction norm: $\|\cdot\|_{\mu,p}$.

**Theorem A.2** (Theorem 2.1 in [10]). *Consider the SA algorithm*

$$x_{k+1} = x_k + \alpha(F(x_k, Y_k) - x_k). \tag{7}$$

*Suppose that*

*(1) The random process $\{Y_k\}$ is a Markov chain (denoted by $\mathcal{MC}_Y$) with finite state-space $\mathcal{Y}$. In addition, $\{Y_k\}$ has a unique stationary distribution $\kappa_Y$, and there exist $C_1 > 0$ and $\sigma_1 \in (0,1)$ such that $\max_{y \in \mathcal{Y}} \|P^k(y, \cdot) - \kappa_Y(\cdot)\|_{TV} \le C_1 \sigma_1^k$ for all $k \ge 0$.*

*(2) The operator $F : \mathbb{R}^d \times \mathcal{Y} \mapsto \mathbb{R}^d$ satisfies for any $x_1, x_2 \in \mathbb{R}^d$ and $y \in \mathcal{Y}$*

    *(a) $\|F(x_1, y) - F(x_2, y)\|_{\mu,p} \le a_1 \|x_1 - x_2\|_{\mu,p}$, where $a_1 > 0$ is a constant,*
    *(b) $\|F(\mathbf{0}, y)\|_{\mu,p} \le b_1$, where $b_1 > 0$ is a constant.*

*(3) The expected operator $\bar{F} : \mathbb{R}^d \mapsto \mathbb{R}^d$ defined by $\bar{F}(x) = \mathbb{E}_{Y \sim \kappa_Y}[F(x, Y)]$ satisfies $\bar{F}(x^*) = x^*$, and is a contraction mapping with respect to $\|\cdot\|_{\mu,p}$, with contraction factor $\gamma_c \in (0,1)$.*

*(4) The constant stepsize $\alpha$ is chosen such that $\alpha t_\alpha(\mathcal{MC}_Y) \le \frac{1-\gamma_c}{228p(a_1+1)^2}$.*

*Then we have for all $k \ge t_\alpha(\mathcal{MC}_Y)$ that*

$$\mathbb{E}[\|x_k - x^*\|_{\mu,p}^2] \le \tilde{c}_1(1 - (1-\gamma_c)\alpha)^{k - t_\alpha(\mathcal{MC}_Y)} + \frac{228p\tilde{c}_2}{(1-\gamma_c)}\alpha t_\alpha(\mathcal{MC}_Y),$$

*where $\tilde{c}_1 = (\|x_0 - x^*\|_{\mu,p} + \|x_0\|_{\mu,p} + b_1/(a_1+1))^2$ and $\tilde{c}_2 = ((a_1+1)\|x^*\|_{\mu,p} + b_1)^2$.*

Proposition A.1 in conjunction with Theorem 2.1 imply that the requirements for applying Theorem A.2 are satisfied. For any $\theta \in (0,1)$, when the constant stepsize $\alpha$ is chosen such that $\alpha\tau_{\alpha,n} \le \frac{\theta\mu_{\min}^{2/p}\omega}{2052pf(\gamma c_{\max})^2(\gamma\rho_{\max}+1)^2}$, we have for any $k \ge \tau_{\alpha,n}$:

$$\mathbb{E}[\|Q_k - Q^{\pi,\rho}\|_{\mu,p}^2] \le \tilde{\zeta}_1(1 - \theta\omega\alpha)^{k - \tau_{\alpha,n}} + \tilde{\zeta}_2 \frac{pf(\gamma c_{\max})^2(\gamma\rho_{\max}+1)^2}{\mu_{\min}^{2/p}\omega}\alpha\tau_{\alpha,n},$$

where $\tilde{\zeta}_1 = (\|Q_0 - Q^{\pi,\rho}\|_{\mu,p} + \|Q_0\|_{\mu,p} + 1)^2$, and $\tilde{\zeta}_2 = 228(3\|Q^{\pi,\rho}\|_{\mu,p} + 1)^2$.

### A.6.2 Proof of Proposition A.1

(1) For any $Q_1, Q_2 \in \mathbb{R}^{|\mathcal{S}||\mathcal{A}|}$ and $y = (s_0, a_0, \cdots, s_n, a_n) \in \mathcal{Y}$, we have

$$\|F(Q_1, s_0, a_0, ..., s_n, a_n) - F(Q_2, s_0, a_0, ..., s_n, a_n)\|_{\mu,p}$$

$$\le \left[\sum_{s,a} \mu(s,a) \left(\mathbb{I}_{\{(s,a)=(s_0,a_0)\}} \sum_{i=0}^{n-1} (\gamma c_{\max})^i(\gamma\rho_{\max}+1)\|Q_1 - Q_2\|_\infty\right)^p\right]^{1/p}$$

$$+ \|Q_1 - Q_2\|_{\mu,p} \qquad\qquad\qquad\text{(Triangle inequality)}$$

$$= f(\gamma c_{\max})(\gamma\rho_{\max}+1)\|Q_1 - Q_2\|_\infty + \|Q_1 - Q_2\|_{\mu,p}.$$

$$\le \frac{2}{\mu_{\min}^{1/p}} f(\gamma c_{\max})(\gamma\rho_{\max}+1)\|Q_1 - Q_2\|_{\mu,p}.$$

Similarly, for any $y = (s_0, a_0, \cdots, s_n, a_n) \in \mathcal{Y}$, we have

$$\|F(\mathbf{0}, s_0, a_0, ..., s_n, a_n)\|_{\mu,p} \le \left[\sum_{s,a} \mu(s,a)\mathbb{I}_{\{(s,a)=(s_0,a_0)\}}\left(\sum_{i=0}^{n-1}(\gamma c_{\max})^i\right)^p\right]^{1/p} \le f(\gamma c_{\max}).$$

(2) Under Assumption 2.1, it is clear that $\{Y_k\}$ has a unique stationary distribution, which we have denoted by $\kappa_Y$, and is given by

$$\kappa_Y(s_0, a_0, ..., s_n, a_n) = \kappa_S(s_0)\left(\prod_{i=0}^{n-1} \pi(a_i|s_i)P_{a_i}(s_i, s_{i+1})\right)\pi(a_n|s_n).$$

Now use the definition of total variation distance, and we have for any $y = (s_0, a_0, ..., s_n, a_n)$ and $k \geq 0$:

$$\|P^{k+n+1}((s_0, a_0, ..., s_n, a_n), \cdot) - \kappa_Y(\cdot)\|_{\text{TV}}$$

$$= \frac{1}{2}\sum_{s_0', a_0', ..., s_n', a_n'}\left|\sum_s P_{a_n}(s_n, s)P_{\pi_b}^k(s, s_0') - \kappa_S(s_0')\right|\left(\prod_{i=0}^{n-1}\pi(a_i'|s_i')P_{a_i'}(s_i', s_{i+1}')\right)\pi(a_n'|s_n')$$

$$= \frac{1}{2}\sum_{s_0'}\left|\sum_s P_{a_n}(s_n, s)P_{\pi_b}^k(s, s_0') - \kappa_S(s_0')\right|$$

$$\leq \frac{1}{2}\sum_{s_0'}\sum_s P_{a_n}(s_n, s)\left|P_{\pi_b}^k(s, s_0') - \kappa_S(s_0')\right|$$

$$= \frac{1}{2}\sum_s P_{a_n}(s_n, s)\sum_{s_0'}\left|P_{\pi_b}^k(s, s_0') - \kappa_S(s_0')\right|$$

$$\leq \frac{1}{2}\sum_s P_{a_n}(s_n, s)\max_{s'}\sum_{s_0'}\left|P_{\pi_b}^k(s', s_0') - \kappa_S(s_0')\right|$$

$$= \max_{s \in \mathcal{S}}\|P_{\pi_b}^k(s, \cdot) - \kappa_S(\cdot)\|_{\text{TV}}$$

$$\leq C\sigma^k.$$

(3) It is clear that $\mathbb{E}_{Y \sim \mathcal{K}_Y}[F(Q, Y)] = \mathcal{K}_{SA}\mathcal{T}_c(\mathcal{H}_\rho(Q) - Q) + Q$, which by definition is equal to $\tilde{\mathcal{B}}_{c,\rho}(Q)$.

## B Connection to Linear SA Involving a Hurwitz Matrix

In view of Proposition 2.2, Algorithm 1 can be alternatively interpreted as a linear SA algorithm for solving the equation $(A - I)Q + b = 0$. In the case where the matrix $A$ is substochastic, our results imply finite-sample bounds for such linear SA algorithm, which is stated in the following.

Let $\{Y_k\}$ be a Markov chain with finite state-space $\mathcal{Y}$ and unique stationary distribution $\kappa_Y$. Let $\tilde{A} : \mathcal{Y} \mapsto \mathbb{R}^{d \times d}$ be a matrix valued function and let $\tilde{b} : \mathcal{Y} \mapsto \mathbb{R}^d$ be a vector valued function. Let $\bar{A} = \mathbb{E}_{Y \sim \kappa_Y}[\tilde{A}(Y)]$ and $\bar{b} = \mathbb{E}_{Y \sim \kappa_Y}[\tilde{b}(Y)]$. Consider the following linear SA algorithm:

$$x_{k+1} = x_k + \alpha((\tilde{A}(Y_k) - I)x_k + \tilde{b}(Y_k)), \tag{8}$$

where $\alpha$ is the constant stepsize. Then, we have the following result.

**Theorem B.1.** *Consider $\{x_k\}$ generated by Algorithm* (8). *Suppose that*

*(1) The Markov chain $\{Y_k\}$ has a unique stationary distribution $\kappa_Y$, and $\max_{y \in \mathcal{Y}}\|P^k(y, \cdot) - \kappa_Y(\cdot)\|_{TV} \leq C'\sigma'^k$ for all $k \geq 0$, where $C' > 0$ and $\sigma' \in (0, 1)$ are constants.*

*(2) There exist $A_{\max}, b_{\max} > 0$ such that $\|\tilde{A}(y)\|_\infty \leq A_{\max}$ and $\|\tilde{b}(y)\|_\infty \leq b_{\max}$ for all $y \in \mathcal{Y}$.*

*(3) The matrix $\bar{A}$ is a sub-stochastic matrix with modulus $\omega' \in (0, 1)$.*

*Then, for any $\theta \in (0, 1)$, when $\alpha$ is chosen such that $\alpha t_\alpha(\mathcal{MC}_Y) \leq \frac{\theta\omega'\mu_{\min}^{2/p}}{228p(A_{\max}+1)^2}$, there exists a weight vector $\mu \in \Delta^d$ satisfying $\mu_{\min} \geq \frac{\omega'(1-\theta)}{(1-\theta\omega')d}$ such that we have for all $k \geq t_\alpha(\mathcal{MC}_Y)$:*

$$\mathbb{E}[\|x_k - x^*\|_{\mu,p}^2] \leq \tilde{c}_1(1 - \theta\omega'\alpha)^{k-t_\alpha(\mathcal{MC}_Y)} + \frac{228p\tilde{c}_2(A_{\max}+1)^2}{\mu_{\min}^{2/p}\theta\omega'}\alpha t_\alpha(\mathcal{MC}_Y),$$

*where $\tilde{c}_1 = (\|x_0 - x^*\|_{\mu,p} + \|x_0\|_{\mu,p} + \frac{b_{\max}\mu_{\min}^{1/p}}{A_{\max}+1})^2$ and $\tilde{c}_2 = (\|x^*\|_{\mu,p} + \frac{b_{\max}\mu_{\min}^{1/p}}{A_{\max}+1})^2$.*

Similarly, Theorem B.1 has the following two corollaries, where we provide finite-sample bounds on $\mathbb{E}[\|x_k - x^*\|_p^2]$ and $\mathbb{E}[\|x_k - x^*\|_\infty^2]$.

**Corollary B.1.1.** *Under the same assumptions as Theorem B.1, we have for all $k \geq t_\alpha(\mathcal{MC}_Y)$:*

$$\mathbb{E}[\|x_k - x^*\|_p^2] \leq \frac{\tilde{c}_1}{\mu_{\min}^{2/p}}(1 - \theta\omega'\alpha)^{k-t_\alpha(\mathcal{MC}_Y)} + \frac{228p\tilde{c}_2(A_{\max}+1)^2}{\mu_{\min}^{4/p}\theta\omega'}\alpha t_\alpha(\mathcal{MC}_Y).$$

**Corollary B.1.2.** *Under the same assumptions as Theorem B.1, we have for all $k \geq t_\alpha(\mathcal{MC}_Y)$:*

$$\mathbb{E}[\|x_k - x^*\|_\infty^2] \leq \tilde{c}_1\sqrt{e}\left(1 - \frac{\omega'\alpha}{2}\right)^{k-t_\alpha(\mathcal{MC}_Y)} + \frac{1824e\log(2d/\omega')\tilde{c}_2(A_{\max}+1)^2}{\omega'}\alpha t_\alpha(\mathcal{MC}_Y).$$

An alternative approach of studying linear SA algorithm with Markovian noise was provided in [29], which established convergence bounds for linear stochastic approximation when the matrix $\bar{A} - I$ is Hurwitz. The bounds in [29] are in terms of solution of the Lyapunov equation:

$$(\bar{A} - I)^\top \Sigma + \Sigma(\bar{A} - I) + I = 0. \tag{9}$$

In particular, the finite-sample bounds depend on the ratio between maximum eigenvalue $\lambda_{\max}$ and minimum eigenvalue $\lambda_{\min}$ of the solution $\Sigma$. In general, it is not clear how this ratio can be evaluated and it is unknown. In the context of off-policy TD algorithms, the dependence on the contraction factor, the variance and the sizes of state-action spaces are hidden in this ratio, and so the trade-offs that we presented in Section 2 are not evident.

In our approach, we overcome this challenge by finding a $\Sigma$ that solves the inequality,

$$(\bar{A} - I)^\top \Sigma + \Sigma(\bar{A} - I) + \eta\Sigma \preceq^2 0 \tag{10}$$

instead of the Lyapunov equation. Here, $\eta > 0$ is a constant. It turns out that this is sufficient to obtain finite-sample bounds. We find such a $\Sigma$ by essentially establishing the weighted $\ell_p$-norm contraction property, in particular, the weighted $\ell_2$-norm contraction property. To see this, observe that when the matrix $\bar{A}$ is a substochastic matrix with a positive modulus $\omega' \in (0, 1)$, Theorem 2.1 implies that $\bar{A}^\top N\bar{A} \preceq (1 - \omega')N$, where $N = \text{diag}(\mu)$. Therefore, we have

$$(1 - \omega')N \succeq (\bar{A} - I + I)^\top N(\bar{A} - I + I) \succeq (\bar{A} - I)^\top N + N(\bar{A} - I) + N,$$

which implies $(\bar{A} - I)^\top N + N(\bar{A} - I) + \omega'N \preceq 0$. Thus, $\Sigma = N$ satisfies (10) with $\eta = \omega'$. When compared to the approach in [29], we trade-off the unknown eigenvalues of the solution to the Lyapunov equation for the weight vector $\mu$. Since we are able to establish a lower bound on $\mu$, we can obtain a sample complexity bound that doesn't involve any unknowns (except $\mathcal{K}_{SA,\min}$, which is inevitable in both approaches). As a result, we are able to fully characterize the impact of the generalized importance sampling ratios in Corollary 2.2.1, and provide insights about the bias-variance trade-offs in multi-step off-policy TD-learning algorithms.

In this section, we present finite-sample bounds for linear stochastic approximation involving a substochastic matrix, whereas [29] considers Hurwitz matrices. However, these results are equivalent because of the following lemma.

**Lemma B.1.** *(1) Suppose $M \in \mathbb{R}^{d \times d}$ is a substochastic matrix with a positive modulus, then the matrix $M'$ defined by $M' = M - I$ is Hurwitz.*

*(2) Suppose $M' \in \mathbb{R}^{d \times d}$ is a Hurwitz matrix, then there exists $\phi \in (0, 1)$ such that the matrix $M = \phi M' + I$ is a contraction mapping with respect to a norm induced by an inner product.*

*Proof.* The proof of Part (1) is straight-forward and we skip it. Part (2) is also not challenging, and we present an overview of the argument. When $M'$ is Hurwitz, all its eigenvalues are located on the open left half of the complex plane. Therefore, there exists $\phi \in (0, 1)$ such that the eigenvalues of $M'$ are within the unit ball centered at $(-1, 0)$ of the complex plane. It follows that $M = \phi M' + I$ has eigenvalues located inside the unit ball centered at the origin of the complex plane. This implies the desired contraction property. $\square$

## C Technical Details in Section 3

### C.1 Proof of Theorem 3.1

Since Vanilla IS is a special case of Algorithm 1, one can directly apply Theorem 2.2 to obtain the finite-sample bound. However, there is one special property of Vanilla IS we can exploit to obtain a tighter finite-sample bound. In particular, consider Proposition A.1 (1) (a). In the case of Vanilla IS, the corresponding Lispchitz constant is $\frac{2}{\mu_{\min}^{1/p}} f(\gamma r_{\max})(\gamma r_{\max} + 1)$. We next show that due to $c(s,a) = \rho(s,a)$ in Vanilla IS, we can use telescoping to improve the Lipschitz constant. Specifically, in Vanilla IS, for any $Q \in \mathbb{R}^{|\mathcal{S}||\mathcal{A}|}$, $y \in \mathcal{Y}$, and $(s,a)$, we have

$$[F(Q,y)](s,a)$$

$$= \mathbb{I}_{\{(s_0,a_0)=(s,a)\}} \sum_{i=0}^{n-1} \gamma^i \prod_{j=1}^{i} c(s_j,a_j)(\mathcal{R}(s_i,a_i) + \gamma c(s_{i+1},a_{i+1})Q(s_{i+1},a_{i+1}) - Q(s_i,a_i)) + Q(s,a)$$

$$= \mathbb{I}_{\{(s_0,a_0)=(s,a)\}} \sum_{i=0}^{n-1} \gamma^i \prod_{j=1}^{i} c(s_j,a_j)\mathcal{R}(s_i,a_i) + \mathbb{I}_{\{(s_0,a_0)=(s,a)\}} \sum_{i=0}^{n-1} \gamma^{i+1} \prod_{j=1}^{i+1} c(s_j,a_j)Q(s_{i+1},a_{i+1})$$

$$- \mathbb{I}_{\{(s_0,a_0)=(s,a)\}} \sum_{i=0}^{n-1} \gamma^i \prod_{j=1}^{i} c(s_j,a_j)Q(s_i,a_i) + Q(s,a)$$

$$= \mathbb{I}_{\{(s_0,a_0)=(s,a)\}} \sum_{i=0}^{n-1} \gamma^i \prod_{j=1}^{i} c(s_j,a_j)\mathcal{R}(s_i,a_i) + \mathbb{I}_{\{(s_0,a_0)=(s,a)\}} \sum_{i=1}^{n} \gamma^i \prod_{j=1}^{i} c(s_j,a_j)Q(s_i,a_i)$$

$$- \mathbb{I}_{\{(s_0,a_0)=(s,a)\}} \sum_{i=0}^{n-1} \gamma^i \prod_{j=1}^{i} c(s_j,a_j)Q(s_i,a_i) + Q(s,a)$$

$$= \mathbb{I}_{\{(s_0,a_0)=(s,a)\}} \sum_{i=0}^{n-1} \gamma^i \prod_{j=1}^{i} c(s_j,a_j)\mathcal{R}(s_i,a_i) + \mathbb{I}_{\{(s_0,a_0)=(s,a)\}}\gamma^n \prod_{j=1}^{n} c(s_j,a_j)Q(s_n,a_n)$$

$$+ (1 - \mathbb{I}_{\{(s_0,a_0)=(s,a)\}})Q(s,a).$$

Therefore, we have for any $Q_1, Q_2 \in \mathbb{R}^{|\mathcal{S}||\mathcal{A}|}$, and $y \in \mathcal{Y}$:

$$\|F(Q_1,y) - F(Q_2,y)\|_{\mu,p}$$

$$\leq \left[ \sum_{s,a} \mu(s,a) \left| \mathbb{I}_{\{(s_0,a_0)=(s,a)\}}\gamma^n \prod_{j=1}^{n} c(s_j,a_j)(Q_1(s_n,a_n) - Q_2(s_n,a_n)) \right|^p \right]^{1/p} + \|Q_1 - Q_2\|_{\mu,p}$$

$$\leq \left[ \sum_{s,a} \mu(s,a) \left| (\gamma r_{\max})^n \|Q_1 - Q_2\|_{\infty} \right|^p \right]^{1/p} + \|Q_1 - Q_2\|_{\mu,p}$$

$$\leq (\gamma r_{\max})^n \|Q_1 - Q_2\|_{\infty} + \|Q_1 - Q_2\|_{\mu,p}$$

$$\leq \frac{(\gamma r_{\max})^n + 1}{\mu_{\min}^{1/p}} \|Q_1 - Q_2\|_{\mu,p}.$$

Using this improved Lipschitz constant and we obtain Theorem 3.1, where the rest of the proof is identical to that of Theorem 2.2.

### C.2 Comparison to the $n$-Step TD-Learning Results in [10]

The sample complexity of on-policy $n$-step TD-learning provided in [10, Corollary 3.3.1.] is

$$\tilde{\mathcal{O}}\left( \frac{n \log^2(1/\epsilon)}{\epsilon^2 \mathcal{K}_{S,\min}^2 (1-\gamma)^2 (1-\gamma^n)^2} \right) \tilde{\mathcal{O}}(|\mathcal{S}|^{1/2}). \tag{11}$$

In this work, by setting $\pi_b = \pi$, Theorem 3.1 implies a sample complexity of

$$\tilde{\mathcal{O}}\left(\frac{n\log^2(1/\epsilon)}{\epsilon^2 \mathcal{K}_{SA,\min}^2 (1-\gamma^n)^2(1-\gamma)^2}\right). \tag{12}$$

These two results have the same dependency on $\epsilon$, $n$, and $1/(1-\gamma)$, but have two differences. First is that in Eq. (11) there is $\mathcal{K}_{S,\min}^{-2}$ while we have $\mathcal{K}_{SA,\min}^{-2}$. This is because we are evaluating the $Q$-function while [10] studies policy evaluation for the $V$-function. Another difference is that in Eq. (11) there is an additional factor of $|\mathcal{S}|^{1/2}$. This is because we find the sample complexity to obtain $\mathbb{E}[\|\cdot\|_\infty] \leq \epsilon$ while [10] finds the sample complexity to achieve $\mathbb{E}[\|\cdot\|_2] \leq \epsilon$.

### C.3 Proof of Theorems 3.2 to 3.5

The results are obtained by directly applying Theorem 2.2.

### C.4 Computing the Sample Complexity of $Q$-Trace from [19]

To compute the sample complexity of the $Q$-trace algorithm from [19], we will adopt the notation from this paper for consistency. In view of [19, Theorem 2.1], to obtain $\mathbb{E}[\|Q_k - Q^{\pi,\rho}\|_\infty] \leq \epsilon$, we need

$$\alpha \sim \mathcal{O}\left(\frac{\epsilon^2}{\log(1/\epsilon)}\right)\tilde{\mathcal{O}}\left(\frac{\mathcal{K}_{SA,\min}^2 f(\gamma D_{c,\min})^2(1-\gamma D_{\rho,\max})^4}{nf(\gamma\bar{c})^2(\gamma\bar{\rho}+1)^2}\right),$$

which implies

$$k \sim \mathcal{O}\left(\frac{\log(1/\epsilon)^2}{\epsilon^2}\right)\tilde{\mathcal{O}}\left(\frac{nf(\gamma\bar{c})^2(\gamma\bar{\rho}+1)^2}{\mathcal{K}_{SA,\min}^3 f(\gamma D_{c,\min})^3(1-\gamma D_{\rho,\max})^5}\right)$$

$$= \tilde{\mathcal{O}}\left(\frac{\log^2(1/\epsilon)nf(\gamma\bar{c})^2(\gamma\bar{\rho}+1)^2}{\epsilon^2\omega^3(1-\gamma D_{\rho,\max})^2}\right).$$