# OpenReview forum: "Finite-Sample Analysis of Off-Policy TD-Learning via Generalized Bellman Operators"
_NeurIPS.cc/2021/Conference — NeurIPS 2021 Poster_

### Official Review · Reviewer_1iu2 · 2021-07-04

**Rating:** 7
**Confidence:** 4

**Summary:**

The authors propose a unified framework for understanding several existing tabular off-policy TD algorithms for policy evaluation. Central to this new framework is a generalized Bellman equation and its contraction. The unified framework provides new insights about the bias-variance trade-off of those existing algorithms, as well as finite sample analysis.

**Limitations And Societal Impact:**

Yes

**Main Review:**

Overall I think this is a solid work and the paper is well written, easy to follow. I checked the proof of Proposition 2.4 in detail, which I believe is the most important ingredient to all the theoretical results in the paper. It looks correct to me. With Proposition 2.4, I do expect all other results to hold.

I have only a few points.
1. In both Line 223 and Line 402, the authors mention the possible extension to linear function approximation. It looks not straightforward to me. With FA, we typically need a projection operator w.r.t. to the norm induced by the sampling distribution (or some other distribution we can sample from). However the generalized Bellman operator is contractive w.r.t. the ergodic distribution of M''. So one possible approach is to design a method to sample according to M''. By the construction of M'', this looks nontrivial. I understand this FA thing is just a possible future, but I might be nice if the authors could discuss this more to give readers more insights.
2. In general I like works unifying existing methods. With this unified framework, is it possible to come up with some new algorithms to optimize the bias variance tradeoff revealed by the analysis? I think by doing so the papers can have more algorithmic contribution.

**Time Spent Reviewing:**

3

---

> ### Author Response · Authors · 2021-08-07
> **Response to Reviewer 1iu2**
>
> (Extend the result to linear function approximation): TD$(0)$ with off-policy sampling and linear function approximation is a typical example of the so-called deadly triad, and the algorithm can in general diverge. The major reason for such divergence is the norm mismatch, i.e., the difference between the projection norm and the contraction norm of the Bellman operator, as pointed out by the reviewer. As a result, the composed operator need not be a contraction. We here present in the following a promising approach to extend our results to the function approximation setting.
>
> To overcome this norm mismatch issue pointed out by the reviewer, we can use $n$-step TD-learning instead of TD$(0)$. The reason is that the $n$-step TD Bellman operator has much smaller contraction factor (e.g. $\gamma^n$ in Vanilla IS) when $n$ is large.  As a result, the small enough contraction factor will overcome the norm mismatch, and the resulting algorithm should converge. Analogous ideas have been exploited in literature with TD$(\lambda)$, where $\lambda$ is chosen close to $1$ to ensure the contraction property [Bertsekas and Yu (2009)][Yu (2012)]. However, [Bertsekas and Yu (2009)][Yu (2012)] only provides asymptotic convergence while our goal is to establish finite-sample bounds.
> Coming back to our paper, the flexibility in choosing the generalized importance sampling ratios and the uniform contraction property of the generalized Bellman operator, we believe, will result in improved finite-sample guarantees in the function approximation setting.
>
> [Bertsekas, D. P., \& Yu, H. (2009). Projected equation methods for approximate solution of large linear systems. Journal of Computational and Applied Mathematics, 227(1), 27-50.]
>
> [Yu, H. (2012). Least squares temporal difference methods: An analysis under general conditions. SIAM Journal on Control and Optimization, 50(6), 3310-3343.]
>
>
> (Optimize the bias variance trade-off): According to the sample complexity bound in Corollary 2.2.1, the bias-variance trade-off is captured by the term $T_3$. Therefore, one can choose the generalized importance sampling ratios with the aim of minimizing $T_3$. We will add a detailed discussion in our revised version. The rule of thumb is that $\gamma c_{\max}$ should be less than $1$, otherwise there will be an exponential term in $T_3$. Based on this rule of thumb, we will include new algorithm designs in the revised version of the paper.

---

### Official Review · Reviewer_USL9 · 2021-07-16

**Rating:** 8
**Confidence:** 4

**Summary:**

This paper proposes a generalized Bellman operator for off-policy methods and provides finite-sample bounds for the associated stochastic approximation algorithm. These bounds can be applied to several previous off-policy TD algorithms such as Retrace and Tree-Backup, which allows one to assess their tradeoffs in the form of a variance term and a contraction factor.


**Limitations And Societal Impact:**

While the limitations are discussed briefly, it would be nice to expand on that: which parts are less satisfying or that could be improved in the present work.

**Main Review:**

Overall, I enjoyed the paper and think it can improve our understanding of off-policy TD methods through a theoretical analysis of sample complexities. The writing was clear and the paper was easy to follow. I particularly appreciated the clear exposition and proof sketches.
I would be interested in seeing the next works in this direction with extensions to function approximation.
I do have some certain clarifications questions written below and would raise my score if these are addressed.

Comments and questions:
- Interesting formulation for the generalized Bellman operator. It seems fruitful to consider those generalizations as a basis for further theoretical analysis.

- The proposed general stochastic approximation update doesn't match the original update for Tree-Backup and Retrace. In particular, there is a importance sampling ratio in the TD-error but, in the original Tree-Backup/Retrace update, there is no such ratio and the target is replaced by \sum_a' pi(a'|s') q(s', a'). This target should intuitively have lower variance than the importance-sampled one and perhaps the bounds can be refined to account for this.
At the very least, it would be important to mention this difference between the original algorithms and the variants currently described in the paper.

- I think the sample bounds established on the various TD methods are interesting and I appreciate that they can meaningfully discriminate between the tradeoffs of each algorithm. In particular, the decomposition into variance and contraction factor is interesting. Pushing this a bit further, can you propose new update rules that improve sample complexity by optimizing these theoretical bounds? I think the theory would be stronger if it could also be to design new algorithms in addition to understanding previous ones.

- As a suggestion to improve clarity, for the individual sample complexities, I suggest omitting the exact iteration bound and specifiyng the step sizes every time since it follows the template of theorem 2.2. I would only include the parameter values to be substituted (c(s,a), \rho(s,a), etc.), along with the value of \omega (the contraction factor) and the sample complexity. It would be nice to additionally include the 'variance' term too. It may be easier to compare algorithms at a glance in this form.

- An explanation of the effect of n (from n-step updates) in the theoretical bounds would be a nice inclusion. As far as I can tell, this only affects the f(.) function which plays a role in both the variance and contraction factor terms.

- A clarification question about proposition 2.4: This proposition guarantees the existence of certain weights \mu for which the Bellman operator is a contraction mapping. Does it matter if this \mu does not correspond to the stationary distribution of any policy? For the standard analyses of on-policy TD, usually the weighting does in fact correspond to the stationary distribution of the behaviour policy.

- Are there any papers analyzing the vanilla IS algorithm and giving finite-sample bounds? I'm a bit surprised there are none since it seems closely related to standard on-policy TD.
- How do these bounds compare to the on-policy TD sample bounds?

Minor comments:
- In several parts, the variable 'a' is overloaded in both the sum and as a argument e.g. line 189 in the definition of D_c, 'a' is used in both the sum and the variable of the function. This would need to be clarified.
- Line 218: The matrix 'A' is referenced but hasn't been introduced up to this point.
- Line 128: Instead of referring to c(.,.) and rho(.,.) both as generalized importance sampling ratios, perhaps the it would be clearer to call c(.,.) something else, such as a 'trace' (following the terminology in the Retrace paper), since it actually consists of a product of the one-step importance sampling ratios.


**Time Spent Reviewing:**

8

---

> ### Author Response · Authors · 2021-08-07
> **Response to Reviewer USL9**
>
> (Original update for Tree-Backup and Retrace): The algorithm we present is the off-policy version of the original TB$(\lambda)$ and Retrace$(\lambda)$, as discussed in [21]. To elaborate, recall that we only have access to the samples $\{(S_k,A_k)\}$ collected under the behavior policy $\pi_b$. However, we need to obtain a sample estimate of the term $E_{\pi}[Q(S,A)|S=s']=\sum_{a'}\pi(a'|s')Q(s',a')$ within the temporal difference (this is an expectation under the target policy $\pi$, and not the behavior policy $\pi_b$). Hence, we need to use $\frac{\pi(A_k|S_k)}{\pi_b(A_k|S_k)}Q(S_k,A_k)$ as our estimate. In this case, since $E_{\pi_b}[\frac{\pi(A|S)}{\pi_b(A|S)}Q(S,A)\mid S]=E_\pi[Q(S,A)\mid S]$, we have an unbiased estimator. We will make this clear in our revised version.
>
> (Can you propose new update rules):
> Our results, especially Theorem 2.1, provide sufficient conditions under which Algorithm 1 has provable finite-sample guarantees, and hence enable us to design new algorithms as suggested by the reviewer.
> In particular,  the bias-variance trade-off is captured by the term $T_3$ in the sample complexity bound in Corollary 2.2.1. Therefore, one can choose the generalized importance sampling ratios with the aim of minimizing $T_3$. The rule of thumb is that $\gamma c_{\max}$ should be less than $1$, otherwise there will be an exponential term in $T_3$. Based on this rule of thumb, we will include a detailed discussion, and new algorithm designs in the revised version of the paper.
>
> (A suggestion to improve clarity): We will incorporate your suggestions into our revised version to improve clarity. In addition, we will add a table to compare the sample complexity bounds of various off-policy TD-learning algorithms studied in this paper.
>
> (The effect of $n$): In addition to appearing in the function $f(\cdot)$ of the term $T_3$ in Corollary 2.2.1, the parameter $n$ also appears linearly in the term $T_2$. Therefore, there is a trade-off in choosing the parameter $n$.  By optimizing the sample complexity bound in terms of $n$, the optimal $n$ is roughly of the size $1/\log(1/\gamma D_{c,\min})$. This is analogous to Corollary 3.3.1 of [9], where on-policy $n$-step TD-learning was studied. We will add this discussion in our revised version.
>
> ($\mu$ does not correspond to the stationary distribution of any policy): It is not problematic if $\mu$ does not correspond to the stationary distribution of any policy.  As long as
> there is a lower bound on the entries of $\mu$, one can use it to convert bounds under $\mu$-weighted norms into bounds under arbitrary weighted norms or even unweighted norms. We present such lower bound of $\mu$ in Theorem 2.1. In Theorem 2.2, we tune the corresponding parameters to obtain the best possible finite-sample bound on $\mathbb{E}[\|Q_k-Q^{\pi,\rho}\|_\infty]$.
>
> However, we argue that one should avoid using weighted norms in presenting the finite-sample bounds, and instead use unweighted norms, in our case, the $\ell_\infty$-norm. For asymptotic convergence, the actual norm used in measuring the distance between the estimate and the true value function is irrelevant because norms are equivalent in finite-dimensional space, and convergence in one specific norm implies convergence in all other norms. However, for finite-sample bounds, using weighted norm in presenting the results does not capture the whole picture because the weight implicitly involves a scaling factor which is related to the size of the state-action space. This is the reason motivating our presentation in terms of the $\ell_\infty$-norm in Theorem 2.2. Please see the first paragraph in Section 2.5 for more details.
>
> (Other papers analyzing vanilla IS algorithm):
> To the best of our knowledge, finite-sample analysis of this type of multi-step off-policy TD-learning algorithms has been performed only recently [8, 9, 18]. For the $Q$-trace algorithm in [18], we can argue that when the truncation levels in $Q$-trace are chosen to be large enough, $Q$-trace reduces to Vanilla IS, and hence the result in [18] implies a finite-sample bound on vanilla IS. However, since we have obtained improved finite-sample bounds for $Q$-trace over [18], see Section 3.1.4, our finite-sample bound on Vanilla IS would also be better. A similar story holds for [8, 9], where they analyze synchronous and asynchronous variants of the $V$-trace algorithm [13], which is used for evaluating the $V$-function as opposed to the $Q$-function in our paper. By setting the truncation levels of $V$-trace to be large enough, $V$-trace recover Vanilla-IS for evaluating the $V$-function. As before, our approach can be used to improve over the resulting finite-sample analysis of $V$-trace in [8,9], in the same manner as we improve the results for the $Q$-trace algorithm. See the last paragraph of Section 3 for more details.
>
> (Comparison to on-policy TD-learning): Finite-sample bounds of on-policy $n$-step TD-learning was given in [9]. Compared to Corollary 3.3.1. of [9], the major difference in our Corollary 2.2.1. is the term $T_3$, which arises because of the generalized importance sampling ratios, and is the unique feature of performing off-policy learning. Please see the paragraph after Corollary 2.2.1. for more details.
>
> (Minor comments): Thank you for your advise, which is very helpful in improving our paper. We will revise our paper based on that.
>
> (Limitations): At this point, we believe that the main limitation of this paper is that it does not apply to function approximation. This direction is our immediate next step. We will elaborate on our plan of extending the result to the linear function approximation setting in the revision (please also see response to Reviewer 1iu2).

---

> > ### Comment · Reviewer_USL9 · 2021-08-20
> > **Response**
> >
> > Thank you for the thorough response. Generally, my concerns have been addressed and I have raised the score to reflect that.
> >
> > I am still unsure about the Tree-Backup and Retrace equivalence though. I understand that we can estimate $E_\pi[Q(S,A)|S=s]$ using importance sampling but, in the original Tree-Backup paper, there is no importance sampling used for this. The expression $\sum_{a'} \pi(a'|s) Q(s,a')$ is directly used as the TD-target since it's available to the agent (see expected SARSA or Tree Backup in Sutton's RL book).
> > I think this is a minor point though.

---

### Official Review · Reviewer_NGvZ · 2021-07-21

**Rating:** 7
**Confidence:** 3

**Summary:**

This paper derives finite-sample bounds for off-policy TD learning algorithms. The main idea is to propose a generalized Bellman operator and rewrite the off-policy TD learning as stochastic approximation algorithms. Then it remains to show under what kind of conditions this generalized Bellman operator is a contraction mapping. The proof technique can be used as a general tool to derive finite sample bounds for existing off-policy learning algorithms.

**Ethical Concerns:**

No ethical issue has been found.

**Limitations And Societal Impact:**

No societal issue has been found.

**Main Review:**

The paper is well written and the problem setups are defined clearly. I really enjoy reading this paper.

The nice thing about the generalized Bellman operator is to provide a unified framework to study the finite sample guarantees of off-policy TD learning algorithms. Also, in the derived bound we can clearly see the bias-variance trade off. Although I am not very familiar with previous theoretical analysis on these algorithms, this does look like a very nice contribution to the literature.

There are some quantities used in the presented results, such as $c_\text{max}, \rho_\text{max}, D_{c, \text{min}}$. It might be good to give some example to show how large these quantities are so that the reader can get some intuitions when the bounds will be large or small.
Furthermore, the paper focus on the policy evaluation problem. How do the results imply for the control setting?

**Time Spent Reviewing:**

3.5

---

> ### Author Response · Authors · 2021-08-06
> **Response to Reviewer NGvZ**
>
> (Quantities such as $c_{\max}$, $\rho_{\max}$, and $D_{c\min}$, etc): We will discuss about these quantities in detail in each off-policy TD-learning algorithm we present. Here as an illustration, we discuss them in the context of Vanilla IS. In the case of Vanilla IS, both $c_{\max}$ and $\rho_{\max}$ are equal to the maximum ratio between the target policy and the behavior policy, i.e., $\max_{s,a}\frac{\pi(a|s)}{\pi_b(a|s)}$, which can be large when the target policy and the behavior policy are very different from each other. In the extreme case where $\pi$ is a deterministic policy and $\pi_b$ is the uniform policy, $c_{\max}$ and $\rho_{\max}$ are both equal to the size of the action space. As for $D_{c,\min}$ and $D_{\rho,\max}$. Observe that in the case of Vanilla IS both $D_c$ and $D_\rho$ are equal to the identity matrix. Therefore we have $D_{c,\min}=D_{\rho,\max}=1$.
>
> (The control setting): TD-learning has been typically used as an intermediate step in solving the control problem in RL. For example, in the popular actor-critic framework, TD-learning is used in the critic to solve the policy evaluation sub-problem. The off-policy TD-learning algorithm we studied in this paper can be incorporated in the actor-critic framework to find the optimal policy. For example, the $Q$-trace algorithm has been used in [18] to form an off-policy variant of the natural actor-critic algorithm. Since we have an improved sample complexity for $Q$-trace over [18], we would expect an improved sample complexity for the off-policy natural actor-critic algorithm as well.

---

### Decision · Program_Chairs · 2021-09-27

**Decision:**

Accept (Poster)

**Comment:**

The reviewers reached a consensus that this paper makes solid contributions. I recommend acceptance. The authors should still carefully revise their paper on the clarity issues raised by the reviewers and explain more about the intuitions for better readablity.